# Prolonged cooling and degassing of Apollo 17 volcanic glasses on the lunar surface

Peng Ni [1] ✉ & Yan Zhan [2] ✉

Studying lunar pyroclastic volcanism provides key insights into the thermo-chemical evolution of the lunar mantle and the Moon's volatile budget. Pyroclastic deposits, although volumetrically minor compared to mare basalts, are derived from deeper, more primitive, and volatile-rich mantle sources. Here we synthesize volatile degassing recorded by Apollo 17 sample 74220, one of most studied pyroclastic deposits. By modeling the volatile degassing of olivine-hosted melt inclusions, melt embayments, and glass beads, we show that volatile depletion of these samples cannot be explained solely by degassing during free-flight, as previously proposed. Instead, the data require extended thermal histories, with beads undergoing slow cooling and prolonged degassing on the lunar surface for years, a scenario further supported by thermal modeling. Our findings suggest that pyroclastic deposits remained active volatile sources well beyond eruption, which is potentially important for sustaining a local transient lunar atmosphere and the long-term volatile cycle of the Moon.

Basaltic volcanism is a fundamental planetary process that had operated on the Moon for billions of years. The timing and extent of lunar basaltic volcanism provide critical information on the internal structure and thermal history of the lunar interior. Besides the effusive eruptions that formed mare basalts on the Moon, fire fountain eruptions are another major style of lunar basaltic volcanism that produced pyroclastic deposits with abundant volcanic glasses[1–3]. More than 100 pyroclastic deposits have been identified using Earth-based remote sensing analyses and Lunar Reconnaissance Orbiter Camera data, two of which cover over 10,000 km² in area[4,5] (Fig. 1).

To distinguish pristine volcanic glasses from impact-generated products, Delano[6] proposed widely accepted criteria, such as the absence of schlieren and exotic inclusions, chemical homogeneity, high Mg/Al ratios, and Ni concentrations that are uniform within individual deposits but vary systematically with Mg content across deposits. The Apollo 15 green glass clods and Apollo 17 orange glass soil are the most well known for their high abundances of pristine glass beads. However, pristine lunar glasses have been identified in samples from all Apollo missions, which have a similarly wide range of TiO₂ concentrations (0–17 wt%) as mare basalts[6]. Variations in TiO₂ concentration produce systematic color changes in the glass beads, from

green to yellow, orange, red, and black with increasing $TiO_2$ content. The widespread distribution of lunar volcanic glasses and their diverse composition range are strong evidence for the prominence of fire fountain eruptions on the Moon.

Lunar volcanic glasses, particularly the Apollo 15 very-low Ti green beads, are more primitive in chemical composition and therefore serve as better proxies for the lunar mantle than mare basalts[6–8]. Their more primitive nature is further supported by multi-point saturation experiments, which indicate that they were likely sourced from depths of ~300–500 km in the lunar mantle[9–12].

Volcanic glass beads have also recently been reported in the Chang'e 5 lunar soil returned by the Chinese mission. Based on U-Pb dating of the beads, they were found to be exceptionally young, suggesting potential volcanic activities on the Moon as recent as 120 Ma ago[13].

Lunar volcanic glasses have been at the forefront of lunar volatiles studies since their return by the Apollo missions. For example, surface enrichments of moderately volatile elements such as Pb, Zn, Cu, and Ge in the form of sulfides or chlorides have been reported for orange glass beads returned by the Apollo 17 mission[2,14–16]. Starting from the first study in 2008 by Saal et al.[17], lunar pyroclastic glasses played a key

[1]Department of Earth, Planetary, and Space Sciences, University of California, Los Angeles, CA, USA. [2]Department of Earth and Environmental Sciences, The Chinese University of Hong Kong, Shatin, NT, Hong Kong. ✉e-mail: pengni@epss.ucla.edu; yanzhan@cuhk.edu.hk

role in lunar sample study "renaissance" as advanced instrumentation enabled the detection of indigenous water in volcanic glass beads and olivine-hosted melt inclusions[18–22]. These discoveries challenged the previous view of a dry Moon expected from its formation in a high-energy giant impact[23], and has contributed to improving existing models of Moon formation[24,25].

Despite the significance of lunar fire fountain eruptions in understanding lunar mantle chemistry and Moon formation, their origin, eruption mechanism, and thermal history remain poorly understood. The thermal history of lunar volcanic glass beads, in particular, is critical for understanding the environment that fire fountain eruptions occurred, the degassing and ingassing of volatiles[26], and for reconstructing the pre-eruptive contents of volatiles in the lunar magma[17].

Among the lunar volcanic glasses returned so far, the Apollo 17 pyroclastic deposit 74220 has the most complete record of volatile degassing (Fig. 2). It contains abundant glass beads with the entire surface exposed to degassing during the eruption[17,20,24]. In contrast, olivine-hosted melt inclusions found in the same sample better preserve pre-eruptive volatile contents (e.g., $H_2O$, F, Cl, S) due to protection by the host olivine[18,20,24,27]. The sample also contains olivine-

hosted melt embayments that show intermediate degassing due to partial enclosure by the host olivine[20,21,27,28].

In this paper, we aim to reconstruct the degassing and thermal history of the fire fountain eruption that produced the pyroclastic deposits at the rim of the Shorty Crater, where Apollo sample 74220 was collected. Using the geochemical records in all the above three types of specimens, we show that the observed volatile depletion in these fire fountain eruption products would require unrealistically long timescales of degassing if it occurred primarily during free-flight. Instead, the high degrees of volatile loss in orange glass beads and melt embayments can be better explained by a prolonged history of slow cooling and degassing on the lunar surface following the eruption.

## Results and discussion

### Summary of volatile degassing records in Apollo 74220 samples

Measured $H_2O$, F, Cl, and S concentrations in olivine-hosted melt inclusions, melt embayments, and glass beads are generally consistent with their expected increasing degrees of volatile loss (Fig. 3). Olivine-hosted melt inclusions found in Apollo 74220 contain 371–1205 ppm $H_2O$, 26–72 ppm F, 1.3–4.8 ppm Cl, and 302–887 ppm S after correcting for the effect of post-entrapment crystallization (PEC) of olivine. These abundances are distinctly higher than those of melt embayments (6.8–38 ppm $H_2O$, 6.0–23 ppm F, 0.029–0.31 ppm Cl, and 182–344 ppm S; corrected for PEC) and glass beads (0.21–12 ppm $H_2O$, 6.7–22 ppm F, 0.014–0.54 ppm Cl, and 218–412 ppm S; PEC correction not needed). Moreover, the melt inclusion data broadly define a linear correlation between $H_2O$ and S (Fig. 3a), and F and S (Fig. 3b), both of which intercept the origin. The trend is less prominent for Cl–S, possibly due to the larger analytical uncertainties associated with the low Cl concentrations.

The correlation between corrected volatile concentrations in melt inclusions from 74220 likely reflects the stage of magma ascent in the conduit before fragmentation and eruption. During ascent, volatiles exsolve from the magma and partition into the vapor phase, leading to a decrease in the volatile concentrations of the magma and an increase in the vapor/liquid ratio. According to the study by Newcombe et al.[29], vapor saturation of the parental magma of 74220 could have occurred at 1.0–6.3 km in the lunar crust, with the carbon content of the magma being a key control which remains poorly constrained. The concentration correlations for melt inclusions can be qualitatively described using zero-intercept linear trends that pass through the

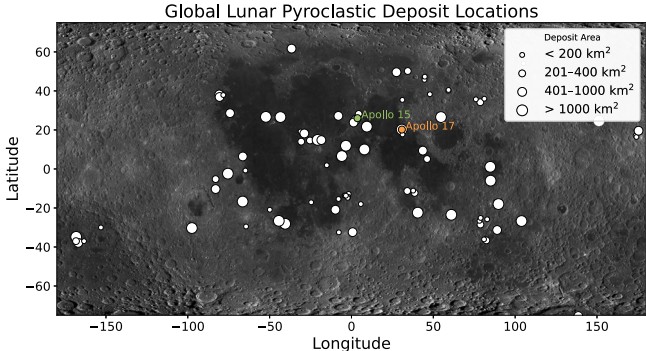

**Global Lunar Pyroclastic Deposit Locations**

**Fig. 1 | Global distribution of identified pyroclastic deposits on the Moon.** Symbol size represents the size of the deposit area. Locations of the Apollo 15 very-low-Ti glasses and Apollo 17 high-Ti glasses are plotted for reference. The lunar map is generated using "Lunar QuickMap". Information of the pyroclastic deposits is from previous studies using Earth-based telescopic observations[4] and Lunar Reconnaissance Orbiter Camera (LROC) data[5].

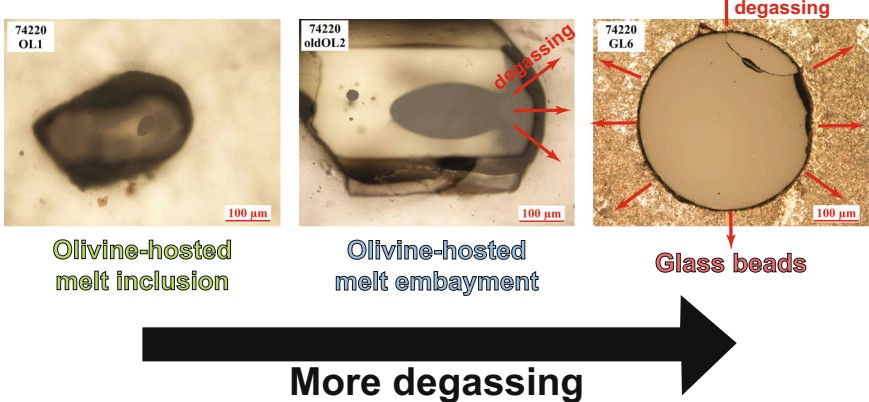

**Fig. 2 | Microscopic images of specimens from Apollo sample 74220 that record different extents of degassing.** Melt inclusions, protected by the host olivine, are the least susceptible to degassing during volcanic eruptions. Embayments are partially open to the olivine surface, allowing for degassing, while glass spherules are fully exposed and therefore completely open to degassing. The olivine grain shown in the middle (74220 OldOL2) contains both melt inclusions and an embayment. Images are from Ni et al.[27] and unpublished thesis work by Ni[28].

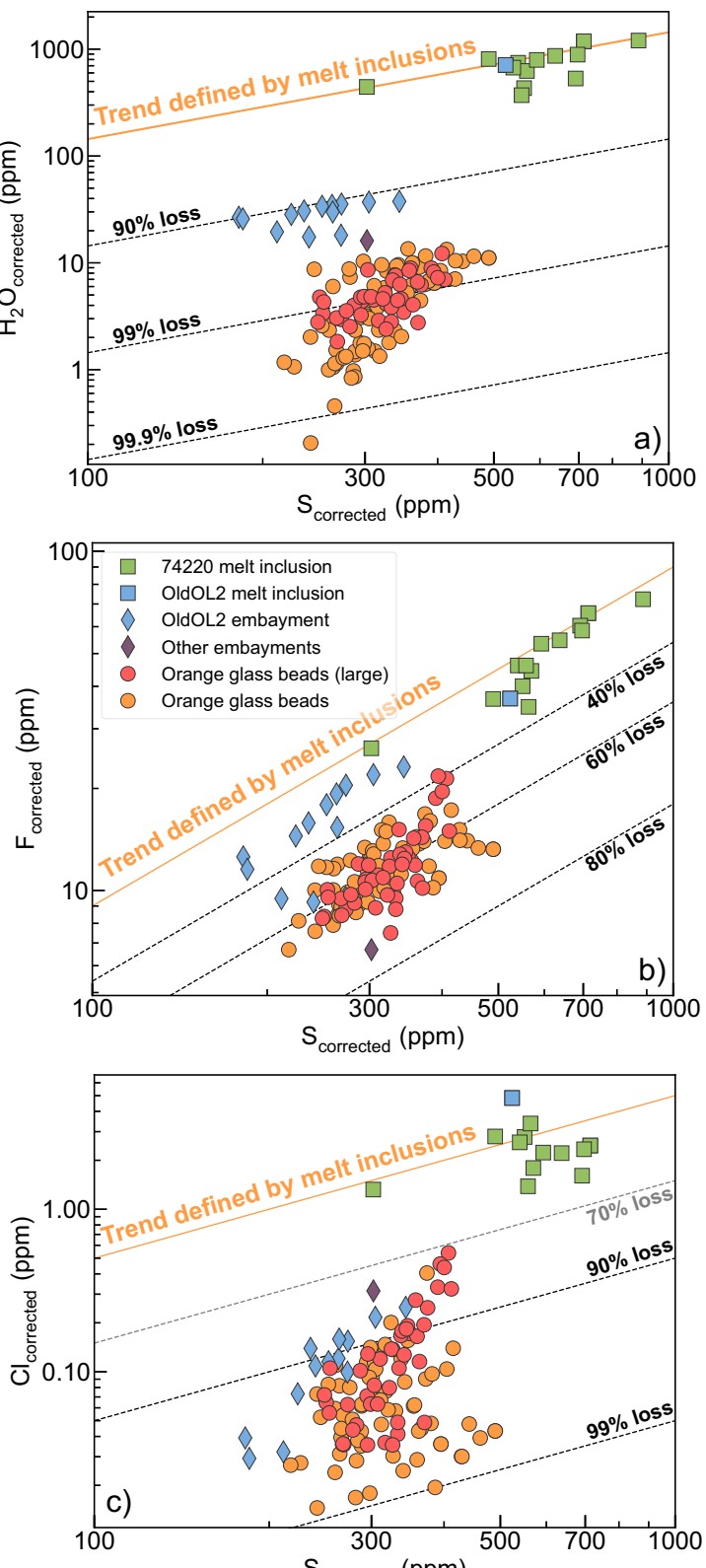

**Fig. 3 | Volatile degassing recorded by specimens from Apollo sample 74220.** Concentrations of **a** $H_2O$, **b** F, and **c** Cl are plotted versus that of S for olivine-hosted melt inclusions, embayments, and glass beads. Volatile concentrations in melt inclusions and melt embayments are corrected for post-entrapment crystallization (PEC) following the procedure in Hauri et al.[18]. The blue diamond symbol represents volatile concentration data measured along the long axis of one large embayment. Glass beads from the "Large mount" in Hauri et al.[18] are colored in red. Data are compiled from literature[17,18,20,21,24,27,28].

upper bounds of the data: $H_2O_{corrected} = 1.44 \times S_{corrected}$, $F_{corrected} = 0.09 \times S_{corrected}$, and $Cl_{corrected} = 0.005 \times S_{corrected}$ (solid orange curves in Fig. 3). Such an approximation assumes that the solubilities of $H_2O$, F, Cl, and S in lunar basalts follow similar power law dependencies on pressure and all approach zero at zero pressure[29]. This assumption is likely inaccurate over a broad pressure range but appears reasonable for the concentration range observed in olivine-hosted melt inclusions (Fig. 3).

Compared with the trend defined by melt inclusions in 74220, orange glass beads from the same sample show stronger depletions in $H_2O/S$, F/S, and Cl/S ratios that can be attributed to post-eruptive degassing. Here, sulfur is used as the denominator because it has the lowest diffusivity among the four volatile components and is, therefore, least susceptible to degassing loss. The greatest depletion is observed in $H_2O/S$ ratios, showing a 98–99.9% decrease in the glass beads relative to the melt inclusions. High levels of depletion, ranging from 70 to 90%, are observed in Cl/S ratios, while lower levels of depletion, between 40 and 70%, are seen in F/S ratios (Fig. 3b, c). Olivine-hosted melt embayments from the same sample show lower degrees of depletion in $H_2O/S$ and Cl/S ratios compared to glass spherules, consistent with partial protection by their host olivine. The depletion in F/S ratios of three of the melt embayments is more significant than that of the glass beads, but the exact reason remains unclear.

One embayment previously reported by Ni et al.[21] was particularly large (Fig. 2, center image). Ellipsoidal in shape and extending ~300 μm along its longest axis, this melt embayment is significantly larger than any other embayment identified in 74220 so far. Therefore, it has the highest likelihood of preserving degassing profiles, which could be used to reconstruct the volcanic eruption process in a manner similar to applications in terrestrial volcanic samples[30,31]. Measurements along the long axis of the exposed embayment show gently decreasing concentration profiles of $H_2O$, F, Cl, and S from the interior toward the opening (Fig. S1, unpublished data from Ni et al.[21]). Notably, the $H_2O$ and Cl concentrations of the melt embayment are significantly lower than those defined by olivine-hosted melt inclusions from 74220, including one from the same host olivine as the large embayment (Fig. 3). Such strong depletions in $H_2O/S$ ( > 90% decrease) and Cl/S ( > 80% decrease) for the melt embayment indicate that diffusive loss of the volatiles was extensive, even at a length scale of ~300 μm.

## Unrealistic timescales needed for in-flight degassing

As summarized above, geochemical records indicate that products of the fire fountain eruption sampled at 74220 experienced at least two stages of degassing. The first stage was decompression degassing during the ascent of the magma in the conduit, as recorded by the olivine-hosted melt inclusions. The second stage began at the moment of magma fragmentation, during which the magma transitioned from a gas-bearing liquid to a gas-dominated phase that carried liquid drops of magma. The rapid expansion of gas drove the eruption, propelling liquid and solid fragments into the space above the crater. The erupted liquid drops of magma became spherical in shape due to surface tension, which eventually cooled down to form the glass beads.

In previous efforts to model volatile degassing in lunar volcanic glass beads, it was typically assumed that volatile loss occurred solely during the free-flight of the beads immediately following the eruption[17,32]. This is equivalent to degassing occurring predominantly during Stage 2 of eruption as described above. The liquid drops that solidified into glass beads could not have formed until magma fragmentation in Stage 2, which supports the above assumption.

With a similar model as previously used in Saal et al.[17], efforts were made to first reproduce the degassing loss of $H_2O$, F, Cl, and S in

orange glass beads relative to the pre-eruptive trends defined by olivine-hosted melt inclusions. The model assumed diffusive loss from a melt sphere:

$$\frac{\partial C_i}{\partial t} = D_i \left( \frac{\partial^2 C_i}{\partial r^2} + \frac{2}{r} \frac{\partial C_i}{\partial r} \right) \Big|_{0 \le r \le a} \tag{1}$$

with the diffusive flux being zero at the center and equal to the evaporative flux at the bead surface:

$$\frac{\partial C_i}{\partial r} \Big|_{r=0} = 0 \tag{2}$$

$$D_i \frac{\partial C_i}{\partial r} \Big|_{r=a} = -v_i C_{i,a} \tag{3}$$

In the above equations, $C_i$ is the concentration of a component in the sphere; $D_i$ is its diffusivity; $v_i$ and $C_{i,a}$ are the evaporation coefficient and the concentration of a component at the surface; and $a$ is the radius of the sphere. Here, the evaporation coefficient $v_i$ represents how fast a component gets lost to the surrounding once it gets transported to the surface of the sphere. The evaporation coefficient of a volatile component is controlled by a range of parameters, such as the equilibrium constant of the evaporation reaction, activity coefficient in the silicate liquid, temperature, and its degree of saturation in the surrounding vapor[33]. These parameters are largely unknown, therefore previous studies typically treated $v_i$ as a free parameter when modeling lunar glass bead degassing[17,32]. It is important to note, however, that the activity coefficient is strongly dependent on temperature. In this study, we instead adopt a dimensionless parameter $L_i = v_i R / D_i$ to connect the evaporation coefficient $v_i$ with diffusivity $D_i$, which evaluates whether the degassing process of a given component is controlled by diffusion (high $L_i$ values) or surface evaporation rate (low $L_i$ values). Using $L_i$ instead of $v_i$ in the model allows us to partially account for the temperature dependence of $v_i$, although its temperature dependence is likely different from that of the diffusivity.

Initial $H_2O$, F, Cl, and S concentrations were assumed to be homogeneous through the beads. Instead of assuming the degassing to occur at a constant temperature as Saal et al.[17], we use an asymptotic cooling model to simulate the cooling history of the beads:

$$T = T_0 / \left(1 + \frac{t}{\tau_c}\right) \tag{4}$$

in which the temperature ($T$) decays exponentially from the initial temperature ($T_0$) with a given cooling time scale ($\tau_c$).

Diffusivities of $H_2O$, F, Cl, and S were calculated as a function of temperature according to Arrhenius's law using experimental data from the literature (Table S1 and Fig. S2). Although the model is capable of calculating the degassing profiles in the beads, volatile concentration data in the literature are typically spot analyses near the center of the beads. Therefore, we calculated the average concentrations of the central 50% (by radius) of the beads to represent the expected measured concentrations from spot analyses. Due to the time-dependence of diffusivity in the model, Eq. (1) cannot be easily solved analytically. We therefore adopted a semi-analytical approach to model the diffusive degassing of $H_2O$, F, Cl, and S for the glass beads. The goal is to obtain a cooling history that can simultaneously explain the degassing loss of all four volatile components. Details of the model are described in "Methods".

Our model targets glass beads that are 50–150 μm in radius, which represent the majority of the orange beads studied for volatile contents in the literature[24,34]. Aside from radius, the most important input parameters of the above diffusion model are the initial temperature, the cooling time scale, and evaporation rates at the melt surface. Our

model aims to constrain the minimum cooling time scales needed to explain the measured $H_2O$, Cl, F, and S concentrations in orange glass beads. For that purpose, we adopted their liquidus of 1603 K as the initial temperature[35]. An initial temperature higher than the liquidus is unlikely, as olivine would be undersaturated in the melt, which contrasts with the observations. In fact, vapor expansion in the conduit is expected to cause adiabatic cooling of the magma during upwelling[36,37]. Using a lower initial temperature in the model would simply result in longer cooling time scales.

Under the above conditions, we conducted a series of sensitivity tests (see Supplementary Discussion 1 for more details) on diffusive degassing of the orange glass beads. The results indicate that a minimum cooling timescale of $10^3$–$10^4$ s (Fig. S3) is required to explain the measured volatile depletions in orange glass beads relative to the most volatile-rich melt inclusion. Among the four volatile components, sulfur has the lowest diffusivity and, therefore, requires the longest time to diffuse away from the center of the sphere. In an extreme scenario, if we assume that the sulfur concentrations in the glass beads were directly inherited from Stage 1 of magma upwelling and no diffusive sulfur loss occurred in Stage 2 during eruption, the model still predicts similar cooling timescales ($10^3$–$10^4$ s) to explain the degassing loss of $H_2O$, F, and Cl (Fig. S4). Under these cooling timescales, the orange beads would take 700–7000 s to cool from their liquidus of 1603 K to the glass transition temperature[38] of 938 K in the asymptotic cooling model. Such a timescale is significantly longer than previously assumed for the degassing of green glass beads[17,32].

Independently, we also modeled the measured $H_2O$, F, Cl, and S concentration profiles in the large ellipsoidal melt embayment in olivine sample 74220 OldOL2, whose longest axis is 286 μm in length. We used a governing equation of one-dimensional diffusion in finite space:

$$\frac{\partial C_i}{\partial t} = D_i \frac{\partial^2 C_i}{\partial x^2}|_{0 \leq x \leq \ell} \qquad (5)$$

in which $\ell$ is the length of the longest axis of the melt embayment.

To model the concentration profiles in the melt embayment, degassing during pre-eruptive ascending (Stage 1) and post-eruptive free-flight (Stage 2) may both be important. This is because, unlike the orange glasses, the olivine-hosted melt embayment formed in the magma conduit at depths. As the magma ascended and underwent decompression, the equilibrium volatile concentration at the outlet of the embayment decreased along the closed-system degassing trend defined by melt inclusions (Fig. 3). During this stage, the temperature is assumed to be constant, and the boundary conditions of the model are defined as:

$$\frac{\partial C_i}{\partial x}|_{x=0} = 0 \qquad (6)$$

and

$$C_i|_{x=\ell} = C_{i,sat}(P) \qquad (7)$$

Here $C_{i,sat}(P)$ is the saturation concentration of a volatile component in the magma as a function of pressure. Assuming a square root law for volatile solubility ($C_{i,sat} = k_i\sqrt{P}$) based on experimental data on $H_2O$ solubility in lunar basalt[29], Eq. (7) converts to $C_i|_{x=\ell} = k_i\sqrt{P}$. Assuming the magma upwells at a constant decompression rate, and would have reached $P = 0$ at the surface within a time of $t_\infty$, Eq. (7) can be further converted to a function of time:

$$C_i|_{x=\ell} = k_i\sqrt{P} = k_i\sqrt{P_0\left(1 - \frac{t}{t_\infty}\right)} = C_{i,0}\sqrt{1 - \frac{t}{t_\infty}} \qquad (8)$$

Here, $C_{i,0}$ is the saturation concentration of a component at the beginning of the model (i.e., $t = 0$). The melt embayment is assumed to have an initial composition that is homogeneous and in equilibrium with the matrix melt. We assume that isothermal decompression ends at $t = t_{end}$ ($t_{end} < t_\infty$), when fragmentation and eruption occur. A large $t_\infty$ value corresponds to slow magma upwelling, whereas a large $t_{end}$ value close to $t_\infty$ indicates that fragmentation occurred very late.

In Stage 2, the host olivine grain erupted into space, exposing the outlet of the melt embayment to open-space degassing. The boundary conditions for the melt embayment in this stage are, therefore, similar to those of the orange beads defined by Eqs. (2) and (3):

$$\frac{\partial C_i}{\partial x}|_{x=0} = 0 \qquad (9)$$

$$D_i \frac{\partial C_i}{\partial x}|_{x=\ell} = -v_i C_{i,\ell} \qquad (10)$$

Here $D_i$, $v_i$, and $C_{i,\ell}$ are the diffusivity, evaporation rate, and concentration at the melt embayment outlet for a volatile component. Similar to our model for the orange glass beads, an asymptotic model (Eq. (4)) was used to simulate the cooling history of the olivine grain in this stage.

Due to the complexity of the two-stage boundary conditions in the melt embayment, Eq. (5) is solved using a semi-analytical approach for the forward modeling of the diffusion profiles. This semi-analytical approach allows us to calculate $H_2O$, F, Cl, and S concentration profiles from any profile at a previous time point (See "Methods"). We adopted a spatial resolution of 1 μm for the model and used variable time steps during cooling to keep diffusivity changes below 1% within each step. The validity of the semi-analytical approach was verified by comparing selected modeled profiles to one-dimensional results produced numerically using COMSOL (Fig. S5). While modeling diffusion in three-dimensional space would theoretically provide greater precision for the melt embayment, it requires significantly more computational resources. A direct comparison of the calculated profiles from our one-dimensional model with two-dimensional axial-symmetric simulations using COMSOL showed that the differences are mostly below 15%, which is reasonable for the purpose of this study (Fig. S5, see Supplementary Discussion 2).

The key parameters of the above two-stage melt embayment degassing model include the upwelling rate (controlled by $t_\infty$ in Eq. (8)), the ending time of Stage 1 ($t_{end}$), the evaporation rate ($v_i$) and the cooling time scale of Stage 2 ($\tau_c$). The distribution of the above parameters was optimized by data assimilation to constrain the cooling time scale and reproduce the observed volatile concentration profiles for the melt embayment in lunar sample 74220. The evaporation rate ($v_i$) for each component was calculated from its diffusivity using given values for dimensionless parameter $L_i$, which is equivalent to the ratio of diffusion time scale ($\tau_{Diff}^i = \ell^2/D_i$) and evaporation time scale ($\tau_{Evap}^i = \ell/v_i$) of component $i$:

$$L_i = \frac{v_i \ell}{D_i} = \frac{\tau_{Diff}^i}{\tau_{Evap}^i} \qquad (11)$$

The value of $L_i$ (or ratio of $\tau_{Diff}^i / \tau_{Evap}^i$) provides a way to directly evaluate whether the degassing process is limited by surface evaporation, diffusion, or both. A high $L_i$ value (e.g., >$10^2$) means that diffusive transport takes significantly more time than surface evaporation, indicating a diffusion-controlled degassing process. A low $L_i$ value (e.g., <$10^{-1}$) leads to the opposite scenario, in which surface evaporation limits degassing. Whereas an intermediate value of $L_i$ indicates that the degassing process is controlled by both surface evaporation and diffusion transport.

**Table 1 | Comparing degassing models for Apollo 15 green bead, Apollo 17 orange beads, and the two-stage degassing model for the olivine-hosted melt embayment**

| Apollo sample # | 15426[a] | 74220 | 74220 |
|---|---|---|---|
| Sample type | Green bead | Orange bead | Melt embayment |
| Radius/length (µm) | 138 | 50 to 150 | 286 |
| $H_2O$ loss | 96% | 97–99.9% | 92% |
| F loss | 32% | 40–70% | 31% |
| Cl loss | 21% | 70-99% | 90% |
| S loss | 0% | 0% | 9% |
| Initial $T$ (K) | 1723 | 1603 | 1603 |
| Cooling time scale (s) | 300[b] | 1,600–32,000[c] | 38,000[c] |

Volatile loss is calculated for the center of the bead or the inner end of the melt embayment.
[a]Data for the Apollo 15 green beads are from Saal et al.[17]
[b]Assuming constant temperature.
[c]Assuming an asymptotic cooling model with a cooling time scale of $\tau_c$: $T = T_0(1 + t/\tau_c)$.

Data assimilation results show that the optimized $L$ values explaining the volatile degassing profiles in the melt embayment OldOL2 cluster around $10^{-1.3}$ for $H_2O$, $10^{-0.9}$ for F, $10^{-0.3}$ for S, and $10^4$ for Cl in the two-stage model (Fig. S5). The optimized $L$ values suggest that Cl degassing in the embayment is diffusion-controlled, $H_2O$ degassing is dominated by surface evaporation, and F and S degassing are influenced by both factors. The $L$ values we obtained are also consistent with the shape of the profiles. The degassing profile is steepest for Cl, consistent with diffusion control, whereas that for $H_2O$ is nearly flat, indicating limited influence from diffusion (Fig. S1).

Most significantly, the optimized cooling time scale ($\tau_c$) of $\sim 10^{4.6}$ s for the melt embayment (Fig. S6) aligns with the higher end of values determined from orange glass beads. Based on our tests, this is the minimum timescale and remains robust despite the choice of parameters for Stage 1 (i.e., $t_\infty$ and $t_{end}$). Such a long time scale is primarily attributed to the need for diffusion to effectively transport F, Cl, and S across 286 µm along the long axis of the melt embayment in Stage 2. We found that degassing in Stage 1 is insignificant in contributing to the eventual degassing in the embayment. This is due to the large departure in $H_2O/S$ and $Cl/S$ ratios from the melt inclusion trend (Fig. 2). Even if S degassing for the melt embayment was completely achieved in Stage 1 (e.g., $t_\infty$ very large), the additional loss of $H_2O$ and Cl requires the dominant degassing process to occur after magma fragmentation.

The cooling timescale of $10^{3.0}$–$10^{4.6}$ s obtained for volcanic beads and olivine-hosted melt embayments from 74220 is significantly longer than the previously assumed free-flight time of 300 s used to model the degassing profiles of $H_2O$, F, Cl, and S in an Apollo 15 green glass bead[17]. As summarized in Table 1, the model results show that orange beads and the melt embayment from 74220 lost similar or greater amounts of $H_2O$, F, Cl, and S compared to the green glass bead from 15426. The length scales are comparable between the green bead (radius of 138 µm) and the orange beads (radii of 50–150 µm), but are significantly greater for the melt embayment (length of 286 µm). The initial temperature, determined from the liquidus of the parental magma, is the biggest difference between models used for degassing in green and orange beads. With a liquidus temperature of ~1723 K[39], the green beads likely erupted at a much higher temperature than the orange beads, which have a liquidus of 1603 K[35]. At such initial temperatures, the diffusivities of $H_2O$, F, Cl, and S in the green bead would have been approximately three times higher than those in the orange beads, enabling significantly more efficient degassing for the green beads. It is also important to note that the thermal history previously used for the Apollo 15 green bead[17] was estimated based on free-flight distances for fire fountain eruption products, rather than being independently constrained.

Our two-stage degassing model requires cooling time scales of 1000–38,000 s in Stage 2 to explain the observed $H_2O$, F, Cl, and S concentrations in orange beads and the olivine-hosted melt embayment from sample 74220. However, such a long cooling timescale is unrealistic, as volcanic beads are estimated to have a maximum free-flight time of approximately 10 min[40]. Since most of them are spherical in shape, they must have cooled below their glass transition temperature of 938 K[38] before landing on the lunar surface. These two constraints limit the cooling timescale of Stage 2 (inflight degassing following eruption) to a few hundred seconds or less, which directly conflicts with our modeling results. Therefore, although a two-stage degassing model is commonly applied to lunar volcanic beads in the literature, it is inadequate to explain the extent of volatile degassing observed in the Apollo 17 high-Ti pyroclastic deposit.

One uncertainty in our two-stage model is the lack of experimental constraints on F, Cl, and S diffusivities in lunar basalts. Previous experiments have shown that $H_2O$ diffusivities in lunar basalts can be a factor of three higher than those in terrestrial basalt at 1330 °C[32]. However, even using F, Cl, and S diffusivities that are five times higher would simply reduce the required cooling timescales by the same factor, which would still be unrealistically long–on the order of thousands of seconds for large glass beads (i.e., >100 µm in radius) and for the melt embayment.

## Prolonged cooling and degassing on the lunar surface

To explain the observed extensive volatile depletion in orange glass beads and olivine-hosted melt embayments from lunar sample 74220, it is essential to consider their degassing history during post-eruptive cooling on the lunar surface–a process that remains largely unexplored. Regolith-like powered materials could have extremely low thermal conductivity under vacuum (on the order of 1 mW m$^{-1}$ K$^{-1}$) as supported by in-situ heat flow measurements on the Moon[41], laboratory measurements of lunar regolith and its simulants[42,43], and equatorial surface temperature data from Diviner[44]. This is due to the highly porous nature of the fine lunar regolith and the absence of gas or water to fill the pores and facilitate heat conduction.

Utilizing a one-dimensional thermal evolution model (see Supplementary Discussion 3), we found that the orange soil, if buried approximately 30 cm beneath the lunar surface (estimated based on the Apollo 17 Preliminary Science Report[45]), could have taken over ten years to cool from its glass transition temperature to ambient conditions (Fig. S7 and Supplementary Discussion 3). Although the detailed thermal history of lunar sample 74220 may depend on factors such as its landing temperature, burial rate, gas pressure, and whether the sampling depth reflects its original burial depth at formation (e.g., due to impact disturbance), the possibility of efficient thermal insulation by an overlying regolith-like layer justifies investigating volatile degassing after landing on the lunar surface.

Using a three-stage thermal history–comprising (1) isothermal upwelling in the magma conduit, (2) rapid cooling during free-flight following the eruption, and (3) slow cooling after landing on the lunar surface (as illustrated in Fig. 4)–and a data assimilation framework (Ensemble Kalman Filter; see "Methods"), we successfully reproduced the $H_2O$, F, Cl, and S concentrations of the orange beads with our diffusion model (Figs. 5a and S8–S11). Note that both a prolonged Stage 2 or Stage 3 may explain the observed high degrees of degassing in the glass beads in the absence of independent time constraints, leading to model degeneracy. In order to estimate the cooling time scale on the lunar surface (Stage 3) needed to explain the observed degassing in $H_2O$, F, Cl, and S, we defined the initial guess of the cooling time scale ($\tau_c$) of Stage 2 to be 100–1000 s. Such a setting is based on: (1) the estimated maximum launching speed of droplets by gas expansion (< 220 m/s) during the eruption, and (2) the maximum ballistic range (< 35 km) estimated for pyroclasts that are 300 µm in diameter[17,46]. The data assimilation results indicate a preferred cooling

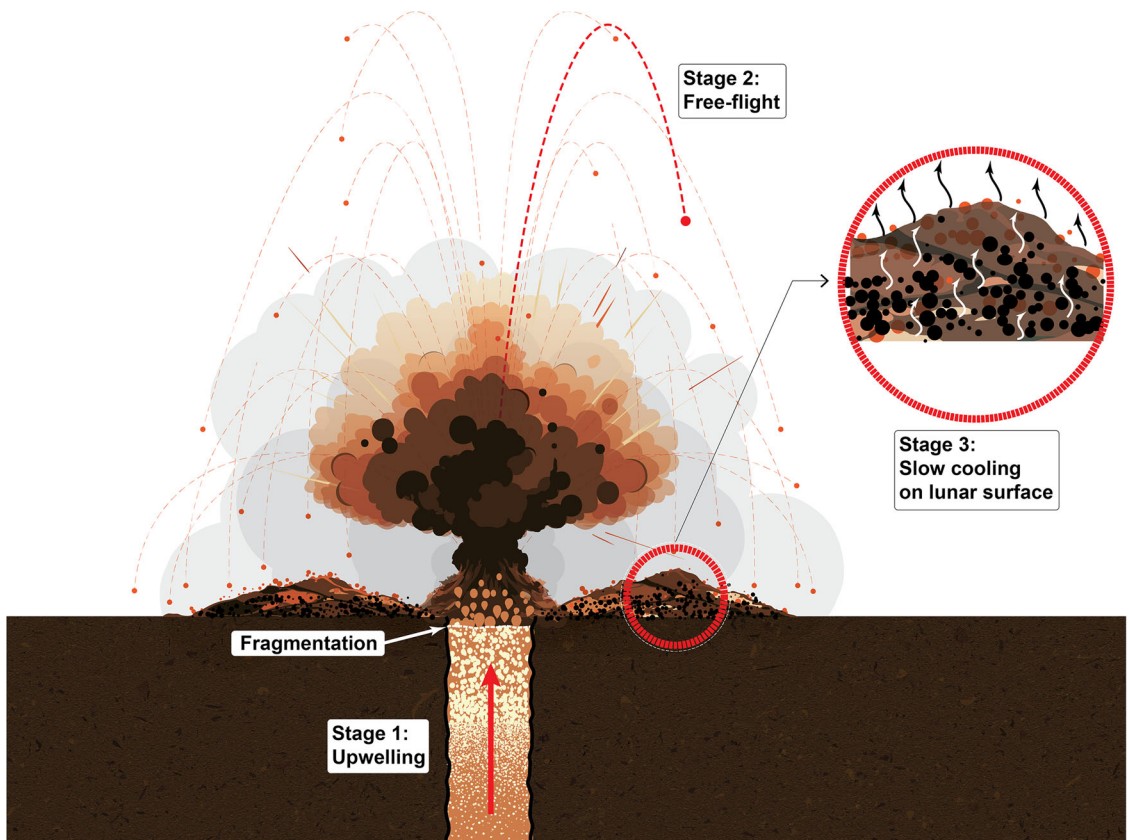

**Fig. 4 | Illustration of the three-stage degassing model for the Apollo 17 fire fountain eruption products.** Stage 1: During magma upwelling, volatiles exsolve from the magma and form bubbles. Degassing during this stage is controlled by decompression, which is recorded by olivine-hosted melt inclusions. Stage 2: Marked by the point of fragmentation, magma quickly converts from a gas-bearing liquid to an expanding gas carrying liquid drops of magma. Droplets of magma get erupted and then fall back to the lunar surface due to gravity, forming glass beads. Stage 3: Glass beads that fell from the eruption quickly accumulate on the lunar surface with other dusts and particles, forming a thick layer of deposit that cools down slowly and continues to degas.

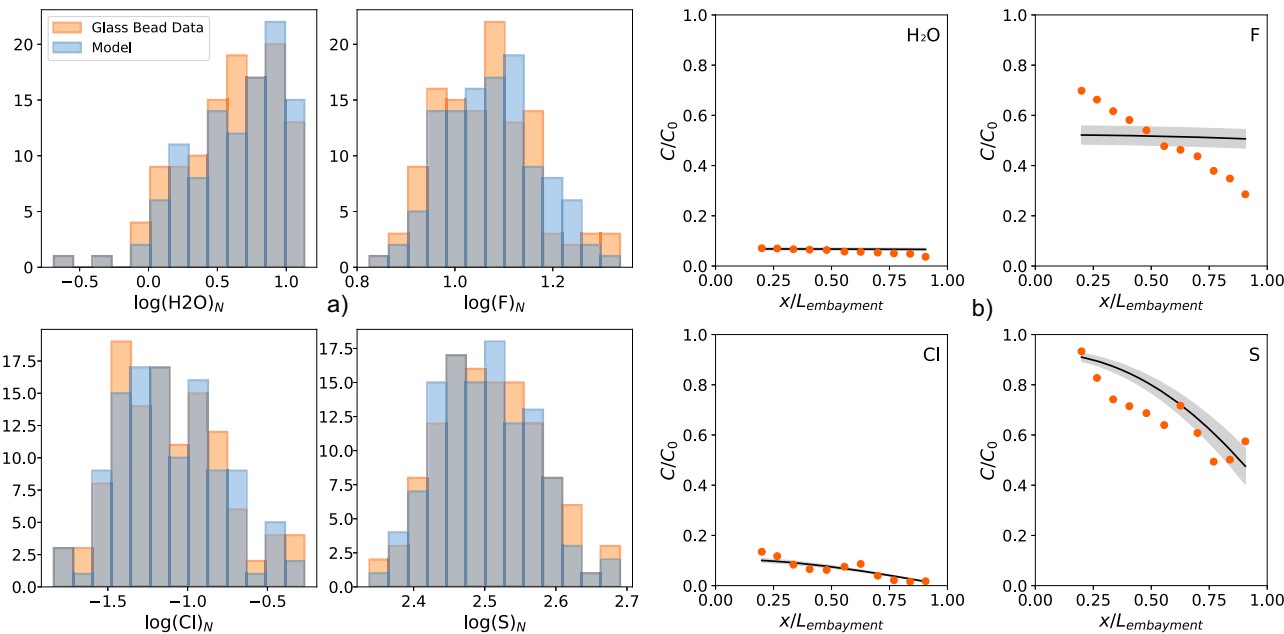

**Fig. 5 | Modeling results generated by data assimilation for the orange beads and the olivine-hosted melt embayment assuming a three-stage degassing history.** Distributions of modeled $H_2O$, F, Cl, and S concentrations in the orange beads are plotted in (**a**) to compare with measurement data. Concentration profiles of the same components modelled for the olivine-hosted melt embayment are plotted in (**b**) with the measurement data. The grey band indicates the model uncertainty propagated from the ±2 standard deviations of the fitted parameters. The model assumes a third stage of solid-state diffusion and degassing for ~$10^8$ s on the lunar surface. The values of model parameters are summarized in Table S2.

timescale of ~250 s during free-flight, corresponding to a cooling rate of 2.2 K/s as the orange beads cool to their glass transition temperature (938 K). This aligns with independent constraints from heat capacity measurements, which estimated a cooling rate of 1.7 K/s across the glass transition for orange glass beads[47].

Data assimilation for the three-stage degassing model shows that the observed $H_2O$, Cl, F, and S degassing in glass beads and the melt embayment are best explained if they had stayed at the glass transition temperature of 938 K for ~$10^8$ s (~3 years) on the lunar surface (Fig. 5). The timescale will be even longer if an asymptotic cooling history is applied to the solid-state degassing in Stage 3. Such a timescale agrees with our thermal modeling results, which show that glass beads buried at 30 cm beneath the surface could take 8 years to cool from 938 to 500 K (Fig. S7).

Due to degeneracy between timescales of Stage 2 and Stage 3, where overlapping contributions to degassing make it difficult to distinguish their individual roles, providing a more quantitatively constrained cooling history for Stage 3 remains challenging. In addition, the lack of volatile concentration profile data in orange glass beads limits the constraints available for our three-stage model. Therefore, our modeling focuses on demonstrating the feasibility of a three-stage thermal history. Further studies of samples from the Apollo 17 pyroclastic deposit, particularly those targeting the third stage of solid-state degassing, will be necessary to better constrain and refine the model.

## Implications of prolonged cooling of Apollo 17 pyroclastic deposit after eruption

Volatiles analyses of sample particles from 74220 have been biased towards orange glass beads because they are glassy and homogeneous, making them easy to analyze with microbeam techniques. In the same pyroclastic deposit, however, there also exist spheres with a progressive range in the degree of devitrification, varying from dark orange to black in color[48]. The texture of devitrification in these beads is dominated by olivine, with ilmenite and other oxides being less common.

It has been proposed in the literature that the devitrification features of the Apollo 17 beads suggest they cooled at rates significantly slower than "free-flight" radiation cooling[38]. Devitrification of the glass spherules in 74220 could have also occurred at sub-solidus temperatures after they landed on the surface of the moon. It is important to note that olivine-hosted melt inclusions and embayments from 74220 all contain dendritic olivine crystals grown from the inclusion–olivine interface (e.g., refs. 18, 27.; also see back-scattered electron images in Fig. S12). This feature differs from the typical concentric growth of olivine associated with PEC, which is indicated by a gradual change in brightness near the melt–olivine interface under back-scattered electron imaging (Fig. S12). We interpret these common devitrification features of black glass beads, melt inclusions, and melt embayments to be all due to slow cooling on the lunar surface.

The negative correlation between cooling rate and burial depth of the pyroclastic deposits (Fig. S7) indicates that earlier-erupted pyroclastic beads cooled more slowly and experienced more extensive devitrification. Slower cooling at greater depths also increases the extent of volatile degassing in these beads (Fig. 4), both for highly volatile elements (e.g., $H_2O$, F, Cl, S) and moderately volatile elements (e.g., Cu, Zn, Pb). The double drive tube sample (74002/74001) taken adjacent to the trench where orange soil 74220 was sampled appear to have features that closely match the above predictions. The top 15 cm of the double drive tube contained a high percentage of orange beads (~70%), consistent with the relatively rapid cooling associated with their shallow burial depth. The percentage of orange beads decreases steadily with depth, with devitrified black beads dominating below ~40 cm[49]. Recent sulfur measurements of glass beads from different layers of the double drill core also yielded higher sulfur concentrations

in shallow samples (231–596 ppm) compared to those at a greater depth (100–244 ppm)[50]. These results are consistent with the expectation that beads buried more deeply underwent more extensive volatile degassing. One uncertainty regarding the double drive tube sample (74002/74001) is whether it preserves the original stratigraphy of the pyroclastic deposit. Cosmogenic noble gas measurements of this sample show that glasses at greater depth contain higher concentrations of cosmogenic $^{21}$Ne, $^{81}$Kr, and $^{131}$Xe, potentially indicating an inversion of the stratigraphy caused by the Shorty Crater impact[51,52]. The above complication has to be considered when testing our model using the double drive tube sample 74002/74001.

Our proposed model of slow cooling and post-eruptive degassing also provides an alternative explanation to the "ingassing" of sulfur, copper, and alkalis recently discovered in the orange glass beads from 74220[26]. In situ measurements using electron microprobe, laser-ablation ICP-MS, and secondary ion mass spectrometry revealed U-shaped profiles of K, Na, and Cu across the central section of the orange beads, indicating diffusion from the surface to the interior. These increasing concentration profiles from the center to the rim of the beads were interpreted as ingassing following outgassing during the pyroclastic eruption[26]. We propose that such an ingassing process could have also occurred on the lunar surface after eruption, as volatiles degassed from greater depths migrated upward and condensed on the surface of orange beads at shallower depths, leading to ingassing of elements such as S, Cu, and alkalis. Such ingassing likely occurred once the orange-bead layer cooled sufficiently to permit sulfide condensation from vapor supplied by deeper degassing. In other words, the orange beads transitioned from the "degassing" regime to "ingassing" regime while they cooled down slowly on the surface of the Moon. Our preliminary results show that the U-shaped and W-shaped S profiles observed in orange glass beads by Su et al.[53] can be reproduced by solid-state diffusion over approximately one month at 938 K (Fig. S13). The actual duration will be longer, as "ingassing" likely occurred late in Stage 3 at lower temperatures. Such a short timescale is consistent with expectations for the beads that cooled down slowly, allowing a brief transition from "degassing" to "ingassing" before temperatures became too low for further diffusion to occur.

The three-stage model shows that the majority of degassing loss at the center of the orange beads occurred during Stage 3 of the model. Under most conditions, diffusion of the volatile components is merely able to reach the bead center by the end of Stage 2. Additional diffusion loss in Stage 3 is fundamental for effectively decreasing the concentrations of $H_2O$, F, Cl, and S at the center of the glass beads.

The lack of olivine-hosted melt inclusions associated with the Apollo 15 green beads makes it difficult to directly constrain the pre-eruptive volatile concentrations of their parental magma. Therefore, our model cannot be used in a similar way to constrain the thermal history of Apollo 15 green beads. Nonetheless, assuming the green beads underwent three stages of degassing, with the same timescales as the orange beads but starting from a higher initial temperature of 1723 K, $H_2O$ at the center of a 300-μm green bead would have depleted by 4 orders of magnitude on average (Supplementary Discussion 5 and Fig. S14), which is inconsistent with the fact that green beads still contain >10 of ppm $H_2O$. We suspect that the Apollo 15 green beads may have cooled down more rapidly than the Apollo 17 orange glass beads during Stage 2 and Stage 3 of degassing.

It is important to note that the thermal history of volcanic glass beads may vary significantly depending on their landing distance from the vent and the intensity of the eruption. Glass beads that fall closer to the vent are likely to experience higher deposition rates, leading to more rapid burial and consequently slower cooling. Similarly, glass beads formed during episodes of high eruption volume may also accumulate more rapidly, resulting in effective insulation and slower cooling. Therefore, interpretation of the thermal history and degassing

records of volcanic glass beads must be placed within a broader spatial and temporal context. From this perspective, future exploration of pyroclastic deposits with undisturbed depositional structures and well-preserved vents–such as the one in the Schrodinger crater[54]–may be critical for building a more complete understanding of this important form of volcanism on the Moon.

In addition to its implications for the mechanism and history of lunar fire fountain eruptions, prolonged cooling and degassing of pyroclastic deposits on the lunar surface may also provide a mechanism for supplying volatiles well beyond the timescales of fire fountain eruptions themselves, potentially sustaining a local transient atmosphere and contributing water and other volatiles to form ice deposits in the permanent shadowed regions of the Moon.

## Methods

### Modeling diffusive volatile loss of volcanic glass beads

The loss of $H_2O$, F, Cl, and S from the volcanic glass beads were modeled assuming radial diffusion in the spherical system:

$$\frac{\partial C_i}{\partial t} = D_i \left( \frac{\partial^2 C_i}{\partial r^2} + \frac{2}{r} \frac{\partial C_i}{\partial r} \right) \Big|_{0 \leq r \leq a} \tag{M1}$$

where $C_i$ is the concentration of a volatile component at radial distance $r$ and time $t$, $D_i$ is the diffusivity of the component, and $a$ is the radius of the sphere. The boundary conditions of Eq. (M1) are defined as:

$$\frac{\partial C_i}{\partial r} \Big|_{r=0} = 0 \tag{M2}$$

$$D_i \frac{\partial C_i}{\partial r} \Big|_{r=a} = -v_i C_{i,a} \tag{M3}$$

Here $v_i$ is the evaporation coefficient of the component and $C_{i,a}$ is the concentration of the component at $r = a$. If $D_i$ is a constant and the sphere has a uniform initial concentration of $C_{i,0}$, the solution to Eq. (M1)[55] is:

$$\frac{C_i}{C_{i,0}} = \frac{2La}{r} \sum_{n=1}^{\infty} \frac{\exp\left(-D\beta_n^2 t/a^2\right)}{\beta_n^2 + L(L-1)} \frac{\sin(\beta_n r/a)}{\sin(\beta_n)} \tag{M4}$$

In the above solution, $L$ is a dimensionless parameter defined as $L = v_i a/D_i$, and $\beta_n$ is the $n_{th}$ root of the transcendental equation:

$$\beta_n \cot \beta_n + L - 1 = 0 \tag{M5}$$

In our model, $D_i$ is not a constant but a temperature-dependent, therefore time-dependent variable that follows the asymptotic thermal history model with an initial temperature of $T_0$ and a cooling timescale of $\tau_c$:

$$T = T_0 / (1 + \frac{t}{\tau_c}) \tag{M6}$$

To calculate the final concentration profile of $C_i$, we derived a general solution to Eq. (M1) for any initial concentration profile of $C_{i,0} = f(r)$:

$$\frac{C_i}{C_{i,0}} = \sum_{n=1}^{\infty} B_n \exp\left(-D\beta_n^2 t/R^2\right) \sin\left(\frac{\beta_n r}{a}\right) \tag{M7}$$

in which $\beta_n$ is still the $n_{th}$ root of the transcendental equation Eq. (M5), while the coefficients $B_n$ are determined by projecting the initial

condition onto the eigenfunctions of $\sin(\beta_n r/a)$:

$$B_n = \frac{1}{r} \frac{\int_0^a \left[ \sin\left(\frac{\beta_n r}{a}\right) f(r) \frac{r}{a} \right] dr}{\frac{1}{2} - \frac{\sin(2\beta_n)}{4\beta_n}} \tag{M8}$$

The initial temperature ($T_0$) is assumed to be the liquidus of the parental liquid of 74220 orange beads ($T_0 = 1603$ K). For a given cooling timescale of $\tau_c$, the thermal history is divided into small segments with a time interval of $\Delta t$. For each time interval, the average temperature is calculated using Eq. (M6), and the temperature-dependent diffusivities of $H_2O$, F, Cl, and S are calculated using experimental data from the literature[32,56,57] that provide the Arrhenius equation (Table S1 and Fig. S2):

$$D_i(T) = D_0 \cdot e^{-\frac{E_a}{R \cdot T}} \tag{M9}$$

with $D_0$ and $E_a$ being the pre-exponential factor and activation energy for each component.

After calculating diffusivities, the concentration profile of each component at the end of the time step is then calculated using Eq. (M7) based on the concentration profile from the previous time step. The calculation proceeds until the diffusion profile converges at lower temperatures. This semi-analytical approach allows for the accommodation of time-dependent changes in diffusivity. It also enables modeling degassing loss across multiple stages of thermal history, as the final profile of one stage can be used as the initial condition for the next. The above semi-analytical approach was benchmarked against numerical results from COMSOL, and consistent results were obtained.

We modeled degassing for glass beads that are 50, 100, and 150 μm in radius ($a$), which represent the majority of orange glass beads studied in literature. The only other unknown parameter for the model is the dimensionless parameter $L_i$, which evaluates the relative rate of surface evaporation and diffusive transport during the degassing process ($L_i = v_i a/D_i$). A high value of $L_i$ indicates high surface evaporation rate relative to diffusion, causing the degassing process to be diffusion-limited. A low value of $L_i$, on the other hand, indicates that the degassing process is limited by surface evaporation. The value of $L_i$ is element-specific and, in theory, should vary with temperature, as evaporation rate ($v_i$) and diffusivity($D_i$) are expected to have different temperature dependencies. However, due to the lack of independent experimental constraints on $v_i$, we simply assume $L_i$ to be constant for each element throughout the cooling history.

### Modeling diffusive volatile loss of the melt embayment

Diffusive loss of $H_2O$, F, Cl, and S in the melt embayment of 74220 OldOL2 was modeled using a similar semi-analytical approach as the glass beads. The model assumes one-dimensional linear diffusion in the embayment that transports volatiles from the interior to the opening for degassing:

$$\frac{\partial C_i}{\partial t} = D_i \frac{\partial^2 C_i}{\partial x^2} \Big|_{0 \leq x \leq \ell} \tag{M10}$$

where $C_i$ is the concentration of a component, $D_i$ is its diffusivity, and $\ell$ is the characteristic length of the embayment. The boundary condition during Stage 1 of magma upwelling is defined as:

$$\frac{\partial C_i}{\partial x} \Big|_{x=0} = 0 \tag{M11}$$

$$C_i|_{x=\ell} = C_{i,0} \sqrt{1 - \frac{t}{t_\infty}} \tag{M12}$$

Here $C_{i,0}$ is the initial concentration of a component in the melt embayment, $t_\infty$ is the amount of time it takes for the magma to reach the surface, assuming linear decompression. We assume that the linear decompression ends at $t = t_{end}$, when fragmentation occurs and explosive eruption starts. Volatile degassing from the melt embayment enters Stage 2 upon eruption, characterized by free-flight and open-space degassing, with a boundary condition at the opening similar to Eq. (M3):

$$D_i \frac{\partial C_i}{\partial x}\big|_{x=\ell} = -v_i C_{i,\ell} \qquad \text{(M13)}$$

In the above equation, $v_i$ is the surface evaporation rate of a component and $C_{i,\ell}$ is the concentration of the component at the opening of the melt embayment ($x = \ell$). During Stage 3 of slow cooling and degassing at the lunar surface, the boundary condition can be described using the same equations for Stage 2 (Eqs. (M11) and (M13)).

Stage 1 of magma upwelling can be assumed to be an isothermal decompression process, making diffusivity $D_i$ time-independent and allowing Eqs. (M10)–(M12) to be solved analytically:

$$C_i = C_{i,0}\sqrt{1 - \frac{t_{end}}{t_\infty}} + \sum_{n=0}^{\infty} \cos\left(\frac{(2n+1)\pi x}{2\ell}\right) e^{-D_i \lambda_n^2 t_{end}}$$
$$\int_0^{t_{end}} e^{D_i \lambda_n^2 t} \frac{4(-1)^{n+1}}{(2n+1)\pi} \frac{df}{dt} dt \qquad \text{(M14)}$$

where $\lambda_n = \frac{(2n+1)\pi}{2l}$, and $\frac{df}{dt} = -\frac{C_{i,0}}{2t_\infty}\sqrt{1 - \frac{t}{t_\infty}}$.

For Stage 2 or Stage 3 with the boundary condition of Eqs. (M11) and (M13), we use a semi-analytical approach that accommodates the time-dependent diffusivity ($D_i$) throughout the thermal history. The thermal history of each stage is divided into time segments that are small enough for the diffusivity to be considered constant within each segment. The initial concentration profile at the beginning of Stage 2 (eruption and in-flight degassing) can be obtained using Eq. (M14), which is a function of $x$. More generally, for any time step during Stage 2 or Stage 3, the initial concentration of a component ($C_i$) is a function of the spatial variable $x$ from the previous time step:

$$C_i\big|_{t=0} = f(x) \qquad \text{(M15)}$$

We derive the solution to Eq. (M10), with the initial condition given by Eq. (M15) and boundary conditions given by Eqs. (M11) and (M13), as:

$$C_i = \sum_{n=1}^{\infty} B_n e^{-(\lambda_n)^2 D_i t/\ell^2} \cos(\lambda_n x/\ell) \qquad \text{(M16)}$$

where

$$B_n = \frac{\int_0^1 f(x)\cos(\lambda_n x/\ell)\,dx}{\frac{1}{2} + \frac{\sin(2\lambda_n)}{4\lambda_n}}, \qquad \text{(M17)}$$

and $\lambda_n$ is the $n_{th}$ root of the transcendental equation with $L_i = v_i \ell / D_i$:

$$\tan\lambda_n = \frac{L_i}{\lambda_n} \qquad \text{(M18)}$$

The above approach allows us to model the diffusive degassing of $H_2O$, F, Cl, and S in the melt embayment. The analytical solutions derived in Eqs. (M14) and (M16) were verified by comparing with numerical simulations using COMSOL. In order to examine whether the one-dimensional approximation is appropriate for the ellipsoidal shape of melt embayment in 74220 OldOL2, we also conducted a finite element simulation for diffusive degassing in a rotationally symmetric

2D system, which closely mimics its original shape (Fig. S5 and Supplementary Discussion 2). We found that the shape of the diffusion profile along the central axis of the 2D model typically differs by less than 15% from that of the 1D semi-analytical model, justifying the use of the 1D model in our study.

We assume a homogeneous initial S concentration of 400 ppm for the embayment and initial $H_2O$, F, Cl concentrations to follow the linear trends defined by olivine-hosted melt inclusions (Fig. 2, $C_{i,H2O} = 1.44 C_{i,S}$, $C_{i,F} = 0.09 C_{i,S}$, and $C_{i,Cl} = 0.005 C_{i,S}$). The thermal history of Stage 2, corresponding to eruption and in-flight degassing, is assumed to follow the asymptotic cooling model in Eq. (M6) with a cooling time scale of $\tau_c$. For simplicity, we simulate the third stage of slow cooling on the lunar surface as lasting a duration of $t_2$ at a constant temperature of 938 K (the glass transition temperature of 74220 orange glass).

Besides diffusivity data from literature[32,56,57], the measured characteristic length ($\ell$) of 286 μm for the melt embayment, $t_{end}$ and $t_\infty$ are the other two parameters required for the Stage 1 of the model (Eq. M14), while $L_i$ (defined as $L_i = v_i \ell / D_i$), $\tau_c$ and $t_2$ are the key parameters for Stage 2 and Stage 3. The unknown parameters are evaluated and optimized using data assimilation techniques, as described below in the next session.

## Data assimilation

The Ensemble Kalman Filter (EnKF) is a statistical tool used to combine observational data with model predictions to estimate the state of a system and its associated parameters. EnKF works by generating an ensemble of possible system states, which are updated iteratively using incoming data[58]. This approach allows for the incorporation of uncertainties in both the model and the observations, making it particularly effective for complex and nonlinear systems[59]. In this study, the EnKF is employed to assimilate measured volatile concentrations of Apollo 74220 pyroclastic deposit samples (e.g., orange glass beads and OldOL2 melt Embayment) into the two-stage or three-stage degassing model, improving the accuracy of predicted volatile evolution.

The EnKF framework begins with the initialization of an ensemble of model states to capture uncertainties in volatile concentrations and parameters. For a given model, an ensemble of 100 members is generated by sampling from uniform distributions based on prior estimates (Table S2). The forward calculation propagates each ensemble member through the selected model.

Data assimilation is performed using the EnKF to update the ensemble based on observational data. The observation vector consists of log-transformed volatile concentrations with an observation error covariance matrix defined using three times the standard deviation (99% confidence interval) of measured concentrations. The root mean square error between the data and model converged quickly after ~10 iterations (e.g., Fig. S9 for the three-stage glass bead degassing model). We conducted a total number of either 15 or 30 iterations to ensure the accuracy of the data assimilations. The final distributions of parameters were verified by examining whether the model effectively captures the central tendencies and ranges of the observed volatile concentrations (e.g., Fig. 5).

In this work, data assimilation was applied to optimize model parameters for three models: (1) a two-stage degassing model for the melt embayment; (2) a three-stage degassing model for orange glass beads; and (3) a three-stage degassing model for the melt embayment. The two-stage degassing model for the melt embayment used Eqs. (M14) and (M16) to calculate volatile concentration profiles at the end of Stage 1 and Stage 2, respectively. The parameters optimized with data assimilation include $t_{end}$, $t_\infty$, $L_i$ and $\tau_c$ (Fig. S6).

The three-stage degassing model for orange glass beads employed a parameter $S_0$ to represent initial S concentration of the bead when they form at the beginning of Stage 2. While other volatiles

($H_2O$, Cl, F) are initialized using ratios relative to $S_0$ based on the melt inclusion trend. This approach ensures a representative spread of initial conditions for subsequent model propagation. Concentration profiles at the end of Stage 2 were calculated using Eq. (M7) with an asymptotic cooling history (Eq. (M6)) from 1603 K (liquidus) to 938 K (glass transition temperature). Stage 3 was subsequently calculated using Eq. (M7), assuming the beads stayed at a constant temperature of 938 K for a duration of $t_2$. Volatile concentrations within the inner 50% of the bead radius were averaged for comparison with the measured data. The parameters optimized with data assimilation include $a$, $S_0$, $L_i$, $\tau_c$ and $t_2$ (Fig. S8 and Table S2).

For simplification, the three-stage degassing model for the melt embayment ignored Stage 1 of decompression and focused on degassing during Stages 2 and 3 instead. This choice was made based on our two-stage model and further tests that showed the volatile loss to be highly limited during Stage 1. Starting from an initially homogeneous profile with 400 ppm S and $H_2O$, F, Cl concentrations defined by the melt inclusion trends (Fig. 2, $C_{i,H2O} = 1.44 C_{i,S}$, $C_{i,F} = 0.09 C_{i,S}$, and $C_{i,Cl} = 0.005 C_{i,S}$), concentration profiles at the end of Stage 2 were calculated using Eq. (M16) with an asymptotic cooling history (Eq. (M6)), followed by application of the same equation at a constant temperature of 938 K for a duration of $t_2$. The parameters optimized with data assimilation include $L_i$, $\tau_c$, and $t_2$ (Table S2).

We found that the optimized $t_2$ for orange glass beads are correlated with the corresponding values of $L_i$ in the three-stage degassing model, owing to the lack of volatile concentration profile data to further constrain these parameters. This correlation is not observed for the melt embayment. Accordingly, our data assimilation focuses on obtaining a consistent thermal history (i.e., similar values of $\tau_c$ and $t_2$) for both the glass beads and the melt embayment.

## Data availability
All data on volatile abundances of melt inclusions, glass beads, and melt embayment used in this study have been summarized as a Source Data file and made available in Figshare (https://doi.org/10.6084/m9.figshare.30077803).

## Code availability
All Python code used for numerical modeling in this work are available on GitHub[60] (https://github.com/geoyanzhan3/ThermoDiffDA; https://doi.org/10.5281/zenodo.18320514).

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

## Acknowledgements

We would like to thank Francis McCubbin and Alberto Saal for helpful discussions on the degassing history of lunar volcanic glass beads. We acknowledge the use of imagery from Lunar QuickMap (https://lunar.quickmap.io), a collaboration between NASA, Arizona State University & Applied Coherent Technology Corp. The project was motivated by a Carnegie Postdoc × Postdoc ($P^2$) seed grant at the Earth and Planets Laboratory, Carnegie Institution for Science. The study was supported in part through the National Science Foundation (NSF) (grant No. EAR-2436722) to P.N., start-up funds provided by UCLA to P.N., and Hong Kong RGC-GRF to Y.Z. (14300325).

## Author contributions

Both authors contributed extensively to the work presented in this paper.

## Competing interests

The authors declare no competing interests.
