## [Transparent Peer Review file · Nature Communications]

Prolonged cooling and degassing of Apollo 17 volcanic glasses on the lunar surface

Corresponding Author: Professor Peng Ni

Version 0:

Reviewer comments:

Reviewer #1

(Remarks to the Author)

Please see attached review.

[Editorial Note: See end of file]

Reviewer #2

(Remarks to the Author)

Reviewer #3

(Remarks to the Author)

Using a combination of original data from an olivine melt embayment and compiled data on volatile element concentrations in lunar orange picritic glass beads, the authors estimate the cooling rate of these Apollo 17 pyroclastic deposits. The authors estimate that the orange picritic glass beads from Apollo 17 cooled over a period of ~17 minutes to ~9 hours which is significantly longer than the ~8 minute cooling period estimated for green picritic glass beads from Apollo 15. The authors conclude that the orange picritic glass beads would require a three-stage cooling and degassing model where prolonged cooling and degassing occurs in piles of pyroclastic material following, magmatic ascent, and ejection from the volcanic vent. This varies from the two-stage cooling and degassing model proposed for the green picritic glass beads from Apollo 15 where the melts primarily cool and degas during the flight away from the volcanic vent following eruption.

The data and models from this study have significant implications for the volatile element (H₂O, Cl, F, and S) concentrations within the moon; however, the authors don't really highlight this in the current form of this manuscript. The way the manuscript currently reads, it seems like the authors attribute the slower cooling rate of orange picritic glass beads to the low thermal conductivity of lunar soils that leads to long term degassing. I believe this overlooks the differences in cooling rate and volatile contents between green and orange glasses. I appreciate that the authors aren't trying to oversell the implications of their findings, but I also feel like they already present the data for a compelling argument about the gas rich nature high-Ti lunar volcanic materials and their mantle sources. The authors should try to focus on answering the following questions: Why would green glasses cool significantly during the free flight stage, but the orange glasses couldn't? The manuscript already notes that orange glasses probably erupt at lower temperatures than green glasses so why would they cool slower? Look at the ratios between slow and fast diffusing volatile elements in green and orange glasses. Green glasses have higher H₂O concentrations and lower F and S concentrations than orange glasses. If the orange glasses cool slower and in-gassing is typically constrained to the surface of the beads, why would the average center composition of orange glass beads have higher F and S concentrations?

The manuscript was well written, and I only found a couple of grammatical errors throughout the text (see line by line comments below). I do think the authors need to move the explanation of some calculation parameters to the first point where they are mentioned (see line by line comments below). The authors also need to clarify some of the modeling assumptions (see line by line comments below). With respect to referencing, I thought the authors did reference the previous literature appropriately, I did point out a couple spots where the authors could reference previous work (see line by line comments below).

I believe the data from previous studies supports the data and interpretations presented within this manuscript. I also believe that the authors are using a good analytical approach to address their questions about the cooling rates of the glass beads, but I did notice some flaws in the diffusion models that need to be addressed prior to publication. For example, I don't think the concentration profiles referenced in Figure 5b currently match the data from the olivine embayment all that well. The authors put a lot of weight on these models explaining the diffusion curves, so this made me question the validity of the authors interpretations and conclusions (see line by line comments for further explanation). I also noticed that the evaporation coefficient used in the model is unconstrained; however, the authors need the evaporation coefficient to calculate the dimensionless diffusion parameter (L). The authors try to work around this using Ensemble Kalman Filter (EnKF) assimilation to determine the optimal dimensionless diffusion parameter, but I found it difficult to assess the sensitivity of the models to this dimensionless diffusion parameter even with the supplemental figures. I would like the authors to show diffusion models that demonstrate the impact of the dimensionless diffusion parameter on the Cl and S models. The best-fit predictions of the dimensionless diffusion parameter for Cl and S have the largest uncertainties based on Table S2 and these elements seem to be the most critical for constraining the maximum cooling rates (i.e., show how much the variance in the best-fit values would affect the predicted cooling rates). Once the authors address these problems, I believe the models they present will support their conclusions without question.

My final assessment of the manuscript is that the manuscript has significant potential for publication in this journal, but it is not publishable in its current state. Once the authors address the comments in this review they should be encouraged to resubmit the manuscript.

Line by line Comments:

Line 39: "are strong evidences" sounds wrong, it should be "are strong evidence"

Line 40: "have recently been also" also sounds wrong, "have also recently been" sounds better.

Lines 51-53: It might be worth saying these elements are "relatively" or "moderately" volatile. While other lunar geochemists will probably understand that these elements are generally volatile at magmatic or impact related temperatures, most people will not. You also refer to Cu, Zn, and Pb as "moderately" volatile elements later in the manuscript.

Figure 3: Increase the size of the text within this figure. It's currently difficult to see what the volatile loss trend lines say without really zooming.

Lines 109-111: Were these trends quantitatively determined using the melt inclusion data? or are these trends qualitative estimates that follow the linear trends defined in this sentence?

Lines 166-169: The evaporation coefficient and its relationship to dimensionless parameter L seems like a significant point of potential error for the models. I looked through the paper for a reference to the evaporation coefficient and found that it is an unconstrained value. This isn't explained until lines 259-276. That explanation should be moved up to this section.

Lines 179-180: The diffusion models from Zhang et al. (2019) (<https://doi.org/10.1016/j.epsl.2019.06.021>) show that H₂O diffuses about an order of magnitude faster in lunar melt compositions than it does in terrestrial basalt compositions. Assuming F, Cl, and S exhibit similar behavior, how might that effect your models?

Lines 225-227: Data from Shishkina et al. (2010) (<https://doi.org/10.1016/j.chemgeo.2010.07.014>) and Botcharnikov et al. (2015) (<https://doi.org/10.1016/j.chemgeo.2015.07.019>) appears to show that volatile saturation and pressure doesn't follow a linear relationship (at least not for H₂O and CO₂). I don't think the authors assumption is all that problematic, given the other model variables that can't be better constrained, but it is worth noting that this assumption isn't completely realistic given our observations in terrestrial systems.

Lines 259-276: Why is the L parameter for Cl in the embayment so different from the L parameter for Cl in the glass beads? Why would Cl degassing be diffusion controlled in the olivine embayment, but evaporation controlled in the glass beads? Cl should be the second fastest diffusing element based on the reported calculation parameters in this study, is there a reasonable explanation as to why it's the only element limited by diffusion or is this a potential problem with the model?

Lines 275-276: This sentence contradicts the data and models presented in Figure 5b. The authors need to check their models and figures to resolve this problem.

Lines 374-377: I don't think the concentration profiles referenced in Figure 5 match the data all that well. The only modeled curve that appears to match the data well is the one for H₂O. Unfortunately, this is the fastest diffusing element, and the trend in the data is effectively flat. Thus, models for this element are only going to provide the authors with a minimum timescale over which diffusion occurred because it has effectively equilibrated across the diffusive length scale being modeled. The S

and F curves aren't even close to matching the data. In the case of S, the data doesn't exhibit a nice diffusion curve so it would be difficult to model that trend so the deviation between the data and the models is understandable. In the case of F and Cl, the data looks fine, but the modeled curves look wrong given the reported diffusivities from Table S1. The current models indicate that F has faster diffusion kinetics than Cl which is inconsistent with the diffusion data shown in Figure 5 and the reported values in Table S1. The authors need to address this problem with new models or explain why the F and Cl models appear to be incorrect.

Lines 435-449: This would be a good point in the manuscript to mention the difference between H₂O vs S and F contents in green glasses compared to the orange glasses (use the supplemental data from Hauri et al. (2015) (<https://doi.org/10.1016/j.epsl.2014.10.053>)). Green glasses have higher H₂O contents than orange glasses on average, but orange glasses have higher S and F contents. S and F should diffuse slowly out of the glasses, but H₂O should diffuse quickly. Are the higher S and F contents of the orange glasses related to the source contents while their lower H₂O contents are related to prolonged cooling? If there's more volatile elements in the source, could that help explain the prolonged cooling history. The authors should also make sure they discuss the new orange glass in-gassing data from Su et al. (2025) (<https://doi.org/10.1016/j.gca.2025.03.026>) since it may support the findings of this study.

Line 439-440: The authors present a model that contradicts this statement in the supplemental data. They show that the green glasses don't fit the three-stage model well and likely cooled faster. I think its reasonable to say that it's less likely that the green glasses experienced significant post flight degassing.

Line 446: "contain 10s of ppm H₂O." should be "contain 10 ppm H₂O."

Lines 473-475: The authors should add a figure to the supplemental information showing how the diffusivity D changes as a function of temperature, using an asymptotic cooling history. This would be helpful for someone trying to comprehend how the cooling rate is impacting the diffusivity D for each element.

Lines 498-507: After reading through the manuscript and supplemental information, the evaporation coefficient and its relationship to dimensionless parameter L seems like a significant point of potential error for the models. The authors noted that the boundary conditions change depending on the stage of degassing in the orange glass cooling models. This change in boundary conditions should affect the evaporation coefficient and as a result the dimensionless parameter L. The authors need to find a better way to demonstrate and explain why the dimensionless parameter L won't cause the models to require unrealistically long cooling histories based on a currently unconstrained evaporation coefficient. Since the authors assume L value is constant, but diffusivity D is changing, they should calculate how the evaporation coefficient u_i would have to change to make this assumption possible. This might be helpful when considering the impact of this assumption on the models.

Figure S1: Flip the x axis in panel b so it's oriented in the same direction as the open end of the OI-embayment. This will make it easier to connect panel a with the trends on panel b.

Table S1: Why don't the D₀ values in this table match any of the studies referenced reported values? Are these averages of multiple of the reported values in those studies? If so that needs to be clarified.

Version 1:

Reviewer comments:

Reviewer #1

(Remarks to the Author)

The revised manuscript is excellent and addresses all of my suggestions. The new modeling of ingassing is a valuable addition. The code (provided on Github) will be a wonderful resource for the community. I fully support publication of the revised manuscript.

(Remarks on code availability)

The code is accessible and clearly documented. The tutorial notebooks provide an excellent introduction to the model.

Reviewer #2

(Remarks to the Author)

(Remarks on code availability)

Code was available and able to be run.

Reviewer #3

(Remarks to the Author)

The authors were very thorough in their response to my comments and the comments of the other two reviewers. I feel like a lot of my previous comments were related to confusion in the way the text read and the way the figures were referenced. The authors have done a great job of fixing this in the revised manuscript. The revised manuscript reads very well and addresses all my previous comments related to the models and calculation parameters. The authors did a good job clarifying the calculation parameters and I feel more confident about the use of the dimensionless parameter L in the calculations and other model assumptions. In conclusion, I believe the authors have adequately addressed all my concerns with the manuscript and the manuscript is publishable in its current state.

Minor Comments:

Figure 5: Fix the spacing between the lower two histograms so the y-axis numbers aren't cut off.

(Remarks on code availability)

Point-by-point responses to reviewer's comments for Manuscript NCOMMS-25-71651 (Revision 1)

Title: Prolonged cooling and degassing of Apollo 17 volcanic glasses on the lunar surface

Authors: Peng Ni and Yan Zhan

The reviewer's comments are in black font while our responses are in blue font.

Reviewer 1&2:

This manuscript constrains a three-stage decompression and cooling history for Apollo 17 volcanic glasses: in the first stage of the model, degassing occurs in response to decompression in the volcanic conduit; in the second stage, the glass beads and embayments undergo diffusive and evaporative loss of volatiles during free flight; and in the third stage of the model, the glasses undergo a final stage of volatile loss during slow cooling on the lunar surface. Volatile concentration gradients in glass beads and melt embayments are best fit by models in which the glasses undergo slow cooling and prolonged degassing on the lunar surface for timescales on the order of years. **This is a novel and interesting finding that leverages previously collected volatiles data with a sophisticated new model.**

The model is carefully constructed. The one-dimensional model for the embayment is checked against a two-dimensional model and is found to agree within 15%. The model uses appropriate parameters and constraints from the literature. In addition to the thorough description of the model provided in the manuscript, the authors state that the code will be provided on Github, which will provide the detail needed for the work to be reproduced. The model results support the conclusions of the manuscript.

We support the publication of this manuscript in Nature Communications. We provide below some suggestions for additional topics of discussion and minor corrections that could strengthen the manuscript.

We sincerely appreciate your support and the insightful suggestions for improving our paper. Our detailed responses are provided below.

Major comments:

1. A linear relationship between H₂O and pressure is assumed, but experimental evidence has shown that H₂O follows a square root relationship with pH₂O (e.g., Newcombe et al. 2017). How might the assumption of a square root relationship affect the results?

This is a good point. At these low pressures, it is possible that all these volatiles follow a square-root law dependence on pressure. In that case, the correlations between the saturated concentrations of H₂O, F, Cl, and S would still be linear because if $C_i = k_i \sqrt{P}$, then $\frac{C_i}{C_j} = k_i/k_j$, which is consistent with the melt inclusion data. Applying the square-root law to volatile solubility is not expected to change our modeling results much, because the majority of volatile loss occurs after magma fragmentation and eruption.

Nonetheless, in order to make our model more robust for community use, we've updated the model to use a square-root law instead of linear law for volatile solubility. The net effect is on Eq. (7) for Stage 1 degassing, which changes from

$$C_i|_{x=\ell} = k_i P = k_i P_0 \left(1 - \frac{t}{t_{\text{inf}}}\right) = C_{i,0} \left(1 - \frac{t}{t_{\text{inf}}}\right)$$

to

$$C_i|_{x=\ell} = k_i \sqrt{P} = k_i \sqrt{P_0 \left(1 - \frac{t}{t_{\text{inf}}}\right)} = C_{i,0} \sqrt{1 - \frac{t}{t_{\text{inf}}}}$$

We've updated the numerical code, re-run all the models, and re-made all the plots in the paper to accommodate the change.

2. Su et al. (2023) suggest that S, Cl and F (which are all enriched on the surfaces of 74220 orange glass beads) are expected to 'ingass' (i.e., diffuse into the beads) as they travel through cool regions of the volcanic plume. The model presented here does not consider ingassing. If ingassing has occurred, the initial stages of ingassing would "flatten" the concentration gradients of H₂O, Cl, S and F, which might cause a bias in the modeled results towards models in which evaporation dominates over diffusion. Notably, this process (referred to as low-temperature rehydration) has also been observed in water concentration gradients across terrestrial melt embayments; e.g. Befus et al. (2024). A discussion of the potential effects of ingassing on the H₂O, Cl, S and F gradients would strengthen the manuscript.

Befus, Kenneth S., et al. "Rehydrated glass embayments record the cooling of a Yellowstone ignimbrite." *Geology* 52.7 (2024): 507-511.

I want to first point out that among S, Cl, F, and H₂O, so far there is only evidence for ingassing of S. Su et al. (2023) reported ingassing of Na, K, and Cu in orange glass beads. A more recent paper by Su et al. (2025) reported ingassing profiles of S in orange glasses. To my best knowledge, nobody has reported ingassing of H₂O, F, or Cl in lunar volcanic glass beads. I have also looked for ingassing profiles in the past, but I've only found ingassing of S, Na, K, and Cu, similar to what Su et al. had reported. There was one recent study on impact glass beads that reported ingassing of H₂O (He et al., 2025). But the source of H₂O was interpreted to be solar wind, and the ingassing profiles were very short. Such ingassing was likely due to brief impact heating, which is irrelevant to volcanic degassing for the orange glass beads.

The occurrence of S ingassing and the absence of H₂O, F, Cl ingassing is likely related to their volatility. It is possible for sulfur to condense from the vapor as sulfides at temperatures much higher than H₂O, F, and Cl. Therefore, even if H₂O, F, and Cl ingassing occurred, their profiles might be much shorter. The rehydration profile in Befus et al. (2024) was interpreted to be caused by meteoric water incorporation into the source magma of the rhyolitic glass. But such water sources are lacking on the Moon.

The reviewers raised an excellent point on how ingassing would "flatten" the concentration gradients of H₂O, F, Cl, and S. If that had happened during Stage 2 (free-flight) as proposed in Su et al. (2025), it would have prohibited further degassing loss of H₂O, F, Cl, and S, requiring even slower cooling (i.e., longer free flight) to explain the

overall degree of volatile depletion at the center of the glass beads or the inner end of the melt embayment. In fact, Su et al. (2025) had to use an artificial sulfur diffusivity **20 times higher** than experimental values to explain the degassing and ingassing of S in orange glasses.

As stated in the discussion session of our paper, an alternative explanation to the S ingassing profiles is that they were produced by solid-state diffusion on the lunar surface (Stage 3). The idea is that glass beads buried at greater depths would cool down slower, continuing to degas and supply sulfur vapor to those buried at shallower depths. When the shallower beads cool down to a certain temperature, sulfides would start to condense on the surface, leading to ingassing of sulfur into the interior of the beads.

In order to better demonstrate the idea, we constructed a forward model for orange glass beads to explain the S ingassing profiles reported by Su et al. (2025). We assumed that toward the end of the degassing in Stage 3, sulfides condense on the surface of orange beads, saturating the surface glass with ~500 ppm of S (depending on the observed S concentration at the rims of those beads) and leading to ingassing of S into those glass beads with different radii. We were able to reproduce the w-shaped and u-shaped sulfur ingassing profiles in Su et al. (2025) within approximately a month of solid-state diffusion (see plot below). It should be possible to reproduce the individual profiles more precisely with our model, but that is beyond the scope of this paper. We've included these updates in the revision to strengthen our manuscript.

Figure R1. Our model provides an alternative explanation to the S ingassing profiles reported in literature. Instead of occurring during flight following the eruption, ingassing could also happen via solid-state diffusion as the volcanic beads slowly cool down on the lunar surface.

3. If it is possible to use the embayment data to report constraints on the decompression rate of the lunar magma (P0/tend?), that would be of great interest.

Unfortunately, we found this to be almost impossible for 74220. The fundamental reason is that most of the volatile loss occurred during Stage 2 (free flight) and Stage 3 (slow cooling on the lunar surface). The volatile profiles at the end of Stage 1 were heavily overprinted by subsequent loss during Stages 2 & 3, leaving poor traces of whether the decompression rate was high or low for magma upwelling. For example, concentrations of H₂O and Cl would have been further depleted by >90% in later stages. Whether the magma upwelling was fast or slow had little effect on the model results. This is discussed on Lines 375 to 378 of the revised manuscript. I see this as a major difference between terrestrial and lunar volcanism in melt embayment studies.

4. At what stage of the decompression and cooling history is post-entrapment crystallization occurring? Is it possible that vapor expansion in the conduit could have led to some adiabatic cooling prior to eruption? E.g., La Spina et al. (2015), Newcombe et al. (2020):

La Spina, G., M. Burton, and M. de'Michieli Vitturi. "Temperature evolution during magma ascent in basaltic effusive eruptions: A numerical application to Stromboli volcano." *Earth and Planetary Science Letters* 426 (2015): 89-100.

Newcombe, Megan E., et al. "Magma pressure-temperature-time paths during mafic explosive eruptions." *Frontiers in Earth Science* 8 (2020): 531911.

This is highly possible. The magma upwelling process is unlikely to be isothermal. Like the reviewers suggest, vapor expansion in the conduit could have led to adiabatic cooling of the magma. The magma would have been subliquidus in temperature when erupted, otherwise it wouldn't have contained olivine grains with melt inclusions. The major element composition of orange glasses also shows an evolution trend (Ni et al., 2017), consistent with magma cooling and fractional crystallization during ascent.

We assumed the magma to be still at its liquidus temperature when erupted, because it helps demonstrate that even with the highest possible initial temperature, volatile degassing loss during free flight alone would still require too long a cooling history to explain the extent of H₂O, F, Cl, and S loss in glass beads and melt embayment.

If a lower, more realistic initial temperature is adopted, the initial diffusivities will be lower, requiring the degassing to occur over a longer timescale. The suggested citations are helpful in demonstrating that our assumed initial temperature is an up limit and have been incorporated into the revised manuscript.

Minor comments:

1. After figure 1, the figure numbers within the text appear to be inaccurate.

Thank you for catching that. This is likely due to the map we added in later as Fig. 1. We've now revised the manuscript to ensure correct figure numbers are cited.

2. Figure 3: The panels and text are small. Could this be a 2x2 tiling with the legend placed in the 4th panel?

We've made this change in the manuscript for the purpose of review. We've also increased the font size to make texts in the figure more readable. Eventually this needs to be a suggestion to the editor for typesetting. The journal might align the three panels together vertically on one of the two columns.

3. Line 7: "thermal evolution of the mantle". Please clarify that you are referring to the *lunar* mantle.

Yes. Fixed for clarification.

4. Line 10: Please clarify "volatile-rich". Do you mean more volatile-rich than the rest of the lunar mantle? Note that scientists from different fields have different views of what constitutes "volatile-rich", so it would be helpful to provide a comparison.

Good point. We meant to say "volatile-rich" relative to mare basalts. We've corrected the text for clarification.

5. Line 34: Please clarify what is meant by "uniform Ni concentrations that correlate with Mg". Do you mean uniform Ni within each pyroclastic deposit?

Yes, volcanic glass beads from each pyroclastic deposit are uniform in Ni, while collectively they show correlation between Ni and Mg that overlap with mare basalts. The phrase has been changed to "Ni concentrations that are uniform within individual deposits vary systematically with Mg content across deposits" for clarification.

6. Line 36: Please emphasize the link between Ti content and glass colors earlier in the text. Glass colors and glass Ti-contents are used interchangeably (e.g. line 35 and 45) and this connection might not be obvious to a reader.

Thanks for the suggestion. A sentence has been added to indicate the connection between Ti-content and glass color.

7. Line 39: "evidences" should be "evidence".

Corrected.

8. Line 56: Perhaps the significance of these glass beads should be emphasized. What is the interest in these samples compared to other lunar lithologies for "understanding lunar mantle chemistry and Moon formation"? This is briefly touched on in the abstract, but may warrant a few more sentences in the introduction.

"Pyroclastic volcanism, although volumetrically minor compared to mare volcanism, is thought to originate from deeper, more primitive sources, and volatile-rich in composition"

Excellent point. Somehow this point was left out when writing the introduction. We've added a couple sentences to elaborate on its importance in lunar mantle chemistry and Moon formation.

Line 73: "contains" is missing an s

Fixed.

Line 100: “volatiles content” shouldn’t be plural

Corrected.

On line 256, what is the difference between t_{inf} and t_{end} ? Please clarify.

t_{inf} in Eq.8 controls the magma upwelling rate, while t_{end} indicates when Stage 1 transitions to Stage 2 of fragmentation and eruption. We realized that t_{end} was not defined from early on. We’ve added a few sentences immediately after Eq. 8 to define and explain t_{end} and t_{inf} (Lines 315-317).

Line 341: Cite the preliminary science report?

Reference added as suggested.

Line 410: “a high percentages” should be “a high percentage”

Corrected.

Line 441: “interpretating” should be “interpreting”

Corrected.

Line 519: Missing “of” between amount and time.

Corrected.

Reviewer 3:

Using a combination of original data from an olivine melt embayment and compiled data on volatile element concentrations in lunar orange picritic glass beads, the authors estimate the cooling rate of these Apollo 17 pyroclastic deposits. The authors estimate that the orange picritic glass beads from Apollo 17 cooled over a period of ~17 minutes to ~9 hours which is significantly longer than the ~8 minute cooling period estimated for green picritic glass beads from Apollo 15. The authors conclude that the orange picritic glass beads would require a three-stage cooling and degassing model where prolonged cooling and degassing occurs in piles of pyroclastic material following, magmatic ascent, and ejection from the volcanic vent. This varies from the two-stage cooling and degassing model proposed for the green picritic glass beads from Apollo 15 where the melts primarily cool and degas during the flight away from the volcanic vent following eruption.

The data and models from this study have significant implications for the volatile element (H₂O, Cl, F, and S) concentrations within the moon; however, the authors don’t really highlight this in the current form of this manuscript. The way the manuscript currently reads, it seems like the authors attribute the slower cooling rate of orange picritic glass beads to the low thermal conductivity of lunar soils that leads to long term degassing. I believe this overlooks the differences in cooling rate and volatile contents between green and orange glasses. I appreciate that the authors aren’t trying to oversell the implications

of their findings, but I also feel like they already present the data for a compelling argument about the gas rich nature high-Ti lunar volcanic materials and their mantle sources. The authors should try to focus on answering the following questions: Why would green glasses cool significantly during the free flight stage, but the orange glasses couldn't? The manuscript already notes that orange glasses probably erupt at lower temperatures than green glasses so why would they cool slower? Look at the ratios between slow and fast diffusing volatile elements in green and orange glasses. Green glasses have higher H₂O concentrations and lower F and S concentrations than orange glasses. If the orange glasses cool slower and in-gassing is typically constrained to the surface of the beads, why would the average center composition of orange glass beads have higher F and S concentrations?

Thanks for the helpful suggestion. This study focuses on Apollo 17 orange glass beads because olivine-hosted melt inclusions and embayments from the same deposit are available and allow us to better constrain the initial conditions required for modeling. We wanted to avoid too much speculation about the green glass beads, for which no melt inclusion data exist to constrain their initial volatile contents.

The volatile concentrations of the green glass beads (H₂O, F, Cl, and S) likely reflect a combination of two factors: (1) higher degrees of partial melting that resulted in lower initial F, Cl, and S contents; and (2) lower degrees of degassing, which led to higher H₂O concentrations compared with the orange glass beads. Additional discussion of the volatile contents of the green glass beads is provided below in our responses to the detailed comments.

Regarding the reviewer's question on the potential causes of differing cooling rates between green glasses and orange glasses, we hope this study encourages broader consideration of the cooling and degassing histories of volcanic glass beads in both spatial and temporal contexts. For example, beads deposited closer to the vent are likely to accumulate more rapidly and cool more slowly, leading to more extensive degassing. Similarly, glass beads deposited during higher-volume eruption episodes are more likely to be buried at higher temperatures and to cool more slowly. Differences in thermal histories between green glasses and orange glasses may therefore reflect variations in distance from the vent, burial depth, and eruption volume. We have added a paragraph to the revised manuscript to discuss this perspective. (Lines 610-619)

The manuscript was well written, and I only found a couple of grammatical errors throughout the text (see line by line comments below). I do think the authors need to move the explanation of some calculation parameters to the first point where they are mentioned (see line by line comments below). The authors also need to clarify some of the modeling assumptions (see line by line comments below). With respect to referencing, I thought the authors did reference the previous literature appropriately, I did point out a couple spots where the authors could reference previous work (see line by line comments below).

I believe the data from previous studies supports the data and interpretations presented within this manuscript. I also believe that the authors are using a good analytical

approach to address their questions about the cooling rates of the glass beads, but I did notice some flaws in the diffusion models that need to be addressed prior to publication. For example, I don't think the concentration profiles referenced in Figure 5b currently match the data from the olivine embayment all that well. The authors put a lot of weight on these models explaining the diffusion curves, so this made me question the validity of the authors' interpretations and conclusions (see line by line comments for further explanation). I also noticed that the evaporation coefficient used in the model is unconstrained; however, the authors need the evaporation coefficient to calculate the dimensionless diffusion parameter (L). The authors try to work around this using Ensemble Kalman Filter (EnKF) assimilation to determine the optimal dimensionless diffusion parameter, but I found it difficult to assess the sensitivity of the models to this dimensionless diffusion parameter even with the supplemental figures. I would like the authors to show diffusion models that demonstrate the impact of the dimensionless diffusion parameter on the Cl and S models. The best-fit predictions of the dimensionless diffusion parameter for Cl and S have the largest uncertainties based on Table S2 and these elements seem to be the most critical for constraining the maximum cooling rates (i.e., show how much the variance in the best-fit values would affect the predicted cooling rates). Once the authors address these problems, I believe the models they present will support their conclusions without question.

Please see our responses to the detailed comments about the definition of parameters and the use of dimensionless parameter L . We've moved up explanations of most parameters in the main text to when they are first introduced.

We understand that the use of dimensionless parameter L is different from previous approaches using evaporation rate v_i and might not be familiar to readers. (Note L is not a diffusion parameter. It is rather the ratio of diffusion timescale over surface evaporation timescale). In fact, these two parameters are inter-calculable for a given length scale: $L_i = v_i R / D_i$. We chose to use L instead of v_i because: 1) v_i is expected to decrease with temperature, using L to connect it with diffusivity D_i , which itself decreases with temperature, helps partly compensate for the temperature dependence of v_i ; 2) L_i provides a more direct way for us to evaluate whether the degassing process is diffusion control or surface evaporation control.

Regarding the impact of the dimensionless parameter L on the model results, we would like to direct our reviewer's attention to Figs. S3 and S4, which present sensitivity tests of the two-stage glass bead degassing model that explicitly demonstrate how variations in L affect the modeling outcomes. At low L values, decreasing L limits the surface evaporation rate, leading to longer timescales for achieving the observed degrees of volatile depletion at the center 50% of the glass beads (marked by light yellow colors in Fig. S2 and S3). In contrast, at high L values, when surface evaporation is rapid, the process becomes diffusion-limited and the timescale no longer decreases with increasing L .

In the revised manuscript, we have added additional discussion of the use of dimensionless parameter L where appropriate to help readers understand this change in model approach relative to literature.

My final assessment of the manuscript is that the manuscript has significant potential for publication in this journal, but it is not publishable in its current state. Once the authors address the comments in this review they should be encouraged to resubmit the manuscript.

Line by line Comments:

Line 39: “are strong evidences” sounds wrong, it should be “are strong evidence”
Corrected.

Line 40: “have recently been also” also sounds wrong, “have also recently been” sounds better.
Corrected.

Lines 51-53: It might be worth saying these elements are “relatively” or “moderately” volatile. While other lunar geochemists will probably understand that these elements are generally volatile at magmatic or impact related temperatures, most people will not. You also refer to Cu, Zn, and Pb as “moderately” volatile elements later in the manuscript.
Good point. Earlier literature referred to all these elements as volatiles. But strictly speaking they should be distinguished from H₂O, F, Cl, and S that are volatile. We’ve changed the wording to “moderately volatile elements”.

Figure 3: Increase the size of the text within this figure. It's currently difficult to see what the volatile loss trend lines say without really zooming.
We’ve increased the font size of all texts in this figure. Thanks for the suggestion.

Lines 109-111: Were these trends quantitatively determined using the melt inclusion data? or are these trends qualitative estimates that follow the linear trends defined in this sentence?
These trends are qualitative estimates that lie on the high end of the melt inclusion data. We decided to use these qualitative estimates instead of linear regression because some of the melt inclusions could have experienced later volatile depletion, such as diffusive H loss. We’ve changed the sentence to say “qualitatively described using zero-intercept linear trends”.

Lines 166-169: The evaporation coefficient and its relationship to dimensionless parameter L seems like a significant point of potential error for the models. I looked through the paper for a reference to the evaporation coefficient and found that it is an unconstrained value. This isn’t explained until lines 259-276. That explanation should be moved up to this section.
Good point. We’ve added a few sentences to further explain L and the evaporation coefficient right after its introduction in the paragraph. In short, L_i is defined as $L_i = v_i R / D_i$. Because both R and D_i are known, L_i and v_i are interchangeable parameters in the model. We choose to fit L_i as a model parameter instead of evaporation rate v_i

because v_i is expected to be a lot more temperature dependent than L_i . Treating v_i as a constant during cooling will likely introduce larger errors on the modeling results.

In addition, the value of L_i gives us a way to directly evaluate whether the degassing process is diffusion-controlled or evaporation-rate-controlled.

Lines 179-180: The diffusion models from Zhang et al. (2019) (<https://doi.org/10.1016/j.epsl.2019.06.021>) show that H₂O diffuses about an order of magnitude faster in lunar melt compositions than it does in terrestrial basalt compositions. Assuming F, Cl, and S exhibit similar behavior, how might that effect your models?

The key question here is whether the degassing observed in orange glass beads and melt embayments can be explained by the traditional two-stage model if volatile diffusivities are significantly higher than those measured experimentally for terrestrial basaltic compositions. In other words, is a three-stage model truly required to explain the observed degassing?

I happened to have recently reviewed a study from the same group on F diffusion in lunar melts, which found that F diffusivity in lunar melts is, on average, approximately 5 times higher than in terrestrial melts. Motivated by this result, we conducted a sensitivity test of the orange glass bead degassing model, assuming that F, Cl, and S diffusivities are five times higher than existing literature values (Fig. R2).

With these elevated diffusivities, the model can explain the volatile loss observed in glass beads with radii of 50 μm . However, explaining degrees of degassing in beads with radii of 100 μm still requires timescales of thousands of seconds, and even longer timescales are needed for beads with radii of 150 μm . Notably, many of the orange glass beads measured were close to 150 μm in radius (see literature data labeled as “large” in Fig. 3). **Therefore, a three-stage-model remains necessary to account for the observed degassing records in 74220 samples.**

Although higher diffusivities for F, Cl, and S would shorten the duration of Stage 3 relative to our current estimates, they do not eliminate the need for this stage. We will make our modeling code openly available on GitHub to facilitate future updates as improved diffusion data on F, Cl, and S become available.

Figure R2: Reproduced Fig. S2 and S3 assuming diffusivities of F, Cl, and S are 5 times higher than in terrestrial basalts. Modeling results that match the observations are marked in light yellow. For glass beads that are 100 μm or 150 μm in radius, thousands of seconds are needed to explain the observed degassing loss of F, Cl, and S even if extremely high L values (therefore evaporation rates) are assumed. This test strengthens our conclusion that a third stage of slow cooling and degassing on lunar surface is necessary to explain the observed volatile depletion in glass beads and melt embayment from 74220.

Lines 225-227: Data from Shishkina et al. (2010)

(<https://doi.org/10.1016/j.chemgeo.2010.07.014>) and Botcharnikov et al. (2015)

(<https://doi.org/10.1016/j.chemgeo.2015.07.019>) appears to show that volatile saturation and pressure doesn't follow a linear relationship (at least not for H₂O and CO₂). I don't think the authors assumption is all that problematic, given the other model variables that can't be better constrained, but it is worth noting that this assumption isn't completely realistic given our observations in terrestrial systems.

This comment is similar to the one given by Reviewers 1&2. Although the impact on our model results is limited as reasoned earlier, we've decided to modify the equations and numerical code to incorporate this change. All results in this paper are re-run with a square root law assumption for solubilities and the difference is minor.

Lines 259-276: Why is the L parameter for Cl in the embayment so different from the L parameter for Cl in the glass beads? Why would Cl degassing be diffusion controlled in the olivine embayment, but evaporation controlled in the glass beads? Cl should be the second fastest diffusing element based on the reported calculation parameters in this study, is there a reasonable explanation as to why it's the only element limited by diffusion or is this a potential problem with the model?

This is likely due to a confusion. The values for L discussed on Lines 259-276 are based on the results from a two-stage model (no third stage of solid-state degassing), while the values for L provided in Table S2 are based on a three-stage model for the glass beads. Therefore, the values for L from these two approaches are not comparable to each other.

Furthermore, in both models Cl tends to have the highest L values among H₂O, F, Cl, and S, which is consistent with the reviewer's comment saying Cl appears to be diffusion limited. The way to understand this result is, low evaporation rate (i.e. low L values) always suppresses degassing, because the volatile component needs to take longer to be lost from the surface melt even if diffusion is efficient enough to transport it to the surface.

The fact that Cl appears to be diffusion-limited in our model is due to the high degrees of depletion in Cl compared to F and S (Fig. 3, main text). In order to explain the high degrees of Cl loss, its evaporation rate needs to be high enough that its degassing doesn't get further limited by surface evaporation (i.e. higher L value). If the L value for Cl was also low, the cooling timescale will be even longer to explain the observed depletions.

A separate line of evidence on L value of the difference components comes from the volatile depletion profiles in the melt embayment. A high L value leads to a degassing scenario that is more diffusion-limited, therefore steeper concentration profile. While a low L value indicates surface evaporation control, leading to flatter profiles (i.e. diffusion is relatively fast to even out the concentration gradient). Among H₂O, F, Cl, and S, Cl has the steepest profile in the melt embayment, consistent with a higher L value.

Lines 275-276: This sentence contradicts the data and models presented in Figure 5b. The authors need to check their models and figures to resolve this problem.

Following the reviewer's suggestions, including this one and a few more below, we've rewritten the discussion on the implications of our modeling results for green glasses.

Lines 374-377: I don't think the concentration profiles referenced in Figure 5 match the data all that well. The only modeled curve that appears to match the data well is the one for H₂O. Unfortunately, this is the fastest diffusing element, and the trend in the data is effectively flat. Thus, models for this element are only going to provide the authors with a minimum timescale over which diffusion occurred because it has effectively equilibrated across the diffusive length scale being modeled. The S and F curves aren't even close to matching the data. In the case of S, the data doesn't exhibit a nice diffusion curve so it would be difficult to model that trend so the deviation between the data and the models is understandable. In the case of F and Cl, the data looks fine, but the modeled curves look wrong given the reported diffusivities from Table S1. The current models indicate that F has faster diffusion kinetics than Cl which is inconsistent with the diffusion data shown in Figure 5 and the reported values in Table S1. The authors need to address this problem with new models or explain why the F and Cl models appear to be incorrect.

Two things need to be clarified in order to address the concerns raised by the reviewer. One is about the reviewer's comment saying, "*The current models indicate that F has faster diffusion kinetics than Cl which is inconsistent with the diffusion data shown in Figure 5 and the reported values in Table S1.*" My guess is that the reviewer saw a flatter F profile than the Cl profile in our model, which led to the confusion that we had used a higher F diffusivity than Cl. However, the shape of the profile is controlled by the L values, not by diffusivities:

High L value -> diffusion control -> steeper profiles
Low L value -> surface evaporation control -> flatter profiles

The modeled F profile is flatter simply because the L value for F is lower than that for Cl, indicating stronger control by the surface evaporation rate. A more effective way to assess the effect of diffusivity is to examine the concentration at the innermost end of the modeled profile. In this case, C/C_0 is approximately 0.63 for F and 0.55 for Cl, consistent with a higher diffusivity for Cl.

Another point I would like to clarify is that the modeled melt embayment profiles in Fig. 5 were generated through forward calculations using the average parameter values derived from the orange glass bead data. We did not perform a separate data assimilation to fit the melt embayment profiles. The purpose of Fig. 5b was to demonstrate how a three-stage model has the potential to simultaneously explain both the glass bead data and the melt embayment data.

Using the three-stage model, we focused on evaluation whether the observed H₂O, F, Cl, and S concentrations at the innermost end of the melt embayment could be reproduced, as these concentrations are more directly controlled by the cooling timescales. The shape of the profiles, in contrast, is primarily influenced by changes in boundary conditions.

Nonetheless, following the reviewer's comment, in the revised manuscript we redid data assimilation for the glass beads and the melt embayment. We focused on achieving high quality fits with similar timescales of both Stage 2 and Stage 3 for the glass beads and the melt embayments. We are able to reproduce the profiles much better this way (Figure 5 of the revised manuscript), with a Stage 2 cooling timescale of ~250 seconds and a Stage 3 duration of 10^8 seconds. The main conclusions of the manuscript remain unchanged.

Lines 435-449: This would be a good point in the manuscript to mention the difference between H₂O vs S and F contents in green glasses compared to the orange glasses (use the supplemental data from Hauri et al. (2015) (<https://doi.org/10.1016/j.epsl.2014.10.053>)). Green glasses have higher H₂O contents than orange glasses on average, but orange glasses have higher S and F contents. S and F should diffuse slowly out of the glasses, but H₂O should diffuse quickly. Are the higher S and F contents of the orange glasses related to the source contents while their lower H₂O contents are related to prolonged cooling? If there's more volatile elements in the source, could that help explain the prolonged cooling history. The authors should also make sure they discuss the new orange glass in-gassing data from Su et al. (2025) (<https://doi.org/10.1016/j.gca.2025.03.026>) since it may support the findings of this study.

Figure R3: Volatile concentration data for green glass beads relative to orange glass beads.

In order to address the reviewer’s question on volatile contents in the green glass beads, we’ve plotted the green glass bead data with those of the orange glass beads for comparison (Fig. R3). The observation that green glasses have relatively high H₂O but lower S and F contents than orange glasses is likely due to the combination of two effects: 1) the parental magma of green glasses was formed by higher degree of partial melting, leading to lower initial volatile contents (because volatiles are incompatible); 2) green glasses experienced lower degrees of volatile degassing, therefore better preserved H₂O than orange glasses.

Fig. R3 further shows that volatile data for the green beads define trends that differ significantly from those of the orange glass beads, suggesting different controlling factors for their degassing processes. For example, both the H₂O-S and Cl-S trends for the green glass beads are flatter than those observed in the orange glass beads. But understanding the degassing trend of green glass beads will likely require substantial effort, given the lack of olivine-hosted melt inclusions to define their initial volatile abundances. This is beyond the scope of the current study. We plan to investigate these differences in future work.

We've mentioned in the manuscript that the S-ingassing data from Su et al. (2025) can also be explained by our three-stage model. In the revised version of the manuscript, we've added additional preliminary modeling to demonstrate this point. We show that the S profiles reported by Su et al. (2025) can be formed in a few weeks on the lunar surface by solid-state diffusion.

Line 439-440: The authors present a model that contradicts this statement in the supplemental data. They show that the green glasses don't fit the three-stage model well and likely cooled faster. I think its reasonable to say that it's less likely that the green glasses experienced significant post flight degassing.

Apologies for the language that confused the reviewer. It was our point that the thermal history of the green glasses was very different.

Line 446: "contain 10s of ppm H2O." should be "contain 10 ppm H2O."
Corrected.

Lines 473-475: The authors should add a figure to the supplemental information showing how the diffusivity D changes as a function of temperature, using an asymptotic cooling history. This would be helpful for someone trying to comprehend how the cooling rate is impacting the diffusivity D for each element.

We've added a figure in the supplemental information on the change of temperature as a function of time, as well as the effect on diffusivities.

Lines 498-507: After reading through the manuscript and supplemental information, the evaporation coefficient and its relationship to dimensionless parameter L seems like a significant point of potential error for the models. The authors noted that the boundary conditions change depending on the stage of degassing in the orange glass cooling models. This change in boundary conditions should affect the evaporation coefficient and as a result the dimensionless parameter L. The authors need to find a better way to demonstrate and explain why the dimensionless parameter L won't cause the models to require unrealistically long cooling histories based on a currently unconstrained evaporation coefficient. Since the authors assume L value is constant, but diffusivity D is changing, they should calculate how the evaporation coefficient v_i would have to change to make this assumption possible. This might be helpful when considering the impact of this assumption on the models.

As explained in response to an earlier comment, we chose to use dimensionless parameter L instead of the evaporation coefficient v_i because v_i is expected to decrease with temperature. In a previous paper (Ni and Shahr, 2023), I showed that the evaporation coefficient for the congruent evaporation of a trace element from a molten sphere can be derived by relating it to the Hertz-Knudsen equation:

$$v = \frac{\alpha_{ec}(1-s)\gamma K M_{melt}}{\rho f(O_2)^{n/4} \sqrt{M}} \sqrt{\frac{1}{2\pi RT}}$$

where α_{ec} is the dimensionless evaporation/condensation coefficient ($0 < \alpha_{ec} < 1$), s is the vapor saturation index, γ is the activity coefficient of the component in silicate liquid, ρ is density of the liquid, n is the number of electron exchange for the evaporation reaction,

K is the equilibrium constant for the evaporation reaction, M and M_{melt} are the molecular masses of the component and the melt, respectively, R is the gas constant and T is temperature.

It is unknown whether the evaporative losses of H₂O, F, Cl, and S are congruent. Moreover, due to the lack of constraints on numerous parameters in the above equation, it is not possible to calculate v directly. However, one point is clear: v is expected to decrease as a function of temperature, owing to the decrease in the equilibrium constant and the additional temperature term highlighted in red.

In previous models, v_i was assumed to be a constant throughout the cooling history. This assumption almost always leads to a scenario of diffusion-limited degassing because v_i is artificially held constant while the diffusivity D_i decreases exponentially with temperature. As a result, $L_i = v_i R / D_i$ increases with decreasing temperature. Our approach of fixing L is still not ideal, because L might also vary with temperature. However, it partially compensates for the temperature dependence of v_i and is better than the previous approach.

The use of dimensionless parameter L also allows us to evaluate the extent of degassing loss under extreme-case scenarios. This is demonstrated by the sensitivity tests for the orange glass beads in Figs. S2 and S3. Even when extremely high L_i values (and thus high evaporation coefficients) are used, the observed volatile depletion in the orange glass beads would still require cooling timescales on the order of thousands of seconds if volatile loss occurred solely during free flight.

In order to address the reviewer's questions, we've added sentences to the main text to further justify our approach of using L instead of v_i in our model.

Figure S1: Flip the x axis in panel b so it's oriented in the same direction as the open end of the Ol-embayment. This will make it easier to connect panel a with the trends on panel b.

Adjusted in the revised manuscript.

Table S1: Why don't the D0 values in this table match any of the studies referenced reported values? Are these averages of multiple of the reported values in those studies? If so that needs to be clarified.

The D0 and Ea values are based on regression of literature data in anhydrous basalt. For example, for H₂O we did a regression of the diffusivities for low-Ti compositions from both Newcombe et al. (2019) and Zhang et al. (2019). H₂O diffusivity data from Zhang et al. (2019) cover very high temperatures between 1480 °C and 1600 °C, while the experiments in Newcombe et al. (2019) were conducted at 1350 °C, closer to the eruption temperature of orange glass. We obtained D0 and Ea for all four components in a similar approach to ensure consistency. In Fig. R3, we also plotted the fitted expressions from original papers in dashed curves for H₂O and S. The differences are relatively minor. Sometimes D0 appears to be quite different from the original publications, but it is compensated by small changes in activation energy Ea. We have now added Fig. R3 to

the supplementary document to clarify on the diffusivity data we used and how D_0 and E_a are obtained.

Figure R3. Diffusion data of H₂O, F, Cl, and S in anhydrous basaltic melts used for constraining D_0 and E_a values in this study (Alletti et al., 2007; Freda et al., 2005; Newcombe et al., 2019; Zhang et al., 2019). Solid curves show linear regression of the data with 95% confidence interval. Original expressions reported by Zhang et al. (2019) for H₂O and Freda et al. (2005) are plotted in dashed curves for comparison.

References:

- Alletti, M., Baker, D.R., Freda, C., 2007. Halogen diffusion in a basaltic melt. *Geochim. Cosmochim. Acta* 71, 3570–3580. <https://doi.org/10.1016/j.gca.2007.04.018>
- Befus, K.S., Thompson, J.O., Allison, C.M., Ruefer, A.C., Manga, M., 2024. Rehydrated glass embayments record the cooling of a Yellowstone ignimbrite. *Geology* 52, 507–511. <https://doi.org/10.1130/G51905.1>
- Freda, C., Baker, D.R., Scarlato, P., 2005. Sulfur diffusion in basaltic melts. *Geochim. Cosmochim. Acta* 69, 5061–5069. <https://doi.org/10.1016/j.gca.2005.02.002>
- He, H., Hu, S., Gao, L., Li, R., Hao, J., Mitchell, R.N., Lu, K., Gao, Y., Li, L., Qiu, M., Zhou, Z., Yang, W., Cai, S., Chen, Y., Jia, L., Li, Q.-L., Hui, H., Lin, Y., Li, X.-H., Wu, F.-Y., 2025. Lunar dichotomy in surface water storage of impact glass beads. *Nat. Commun.* 16, 4971. <https://doi.org/10.1038/s41467-025-60388-y>
- Newcombe, M.E., Beckett, J.R., Baker, M.B., Newman, S., Guan, Y., Eiler, J.M., Stolper, E.M., 2019. Effects of pH₂O, pH₂ and fO₂ on the diffusion of H-bearing species in lunar basaltic liquid and an iron-free basaltic analog at 1 atm. *Geochim. Cosmochim. Acta* 259, 316–343. <https://doi.org/10.1016/j.gca.2019.05.033>

- Ni, P., Shahar, A., 2023. Copper isotope fractionation by diffusion in a basaltic melt. *Earth Planet. Sci. Lett.* 624, 118459. <https://doi.org/10.1016/j.epsl.2023.118459>
- Ni, P., Zhang, Y., Guan, Y., 2017. Volatile loss during homogenization of lunar melt inclusions. *Earth Planet. Sci. Lett.* 478, 214–224.
- Su, X., Zhang, Y., Liu, Y., 2025. Sulfur outgassing and in-gassing in lunar orange glass beads and implications for ³³S “Anomaly” in the Moon. *Geochim. Cosmochim. Acta* 397, 164–175. <https://doi.org/10.1016/j.gca.2025.03.026>
- Su, X., Zhang, Y., Liu, Y., Holder, R.M., 2023. Outgassing and in-gassing of Na, K and Cu in lunar 74220 orange glass beads. *Earth Planet. Sci. Lett.* 602, 117924. <https://doi.org/10.1016/j.epsl.2022.117924>
- Zhang, Li, Guo, X., Li, W.-C., Ding, J., Zhou, D., Zhang, Lingmin, Ni, H., 2019. Reassessment of pre-eruptive water content of lunar volcanic glass based on new data of water diffusivity. *Earth Planet. Sci. Lett.* 522, 40–47. <https://doi.org/10.1016/j.epsl.2019.06.021>

Point-by-point responses to reviewer's comments for Manuscript NCOMMS-25-71651 (Acceptance)

Title: Prolonged cooling and degassing of Apollo 17 volcanic glasses on the lunar surface

Authors: Peng Ni and Yan Zhan

The reviewers did not raise further comments besides a fix of format for Figure 5. The reviewer's comments are in black font while our responses are in blue font.

REVIEWERS' COMMENTS

Reviewer #1 (Remarks to the Author):

The revised manuscript is excellent and addresses all of my suggestions. The new modeling of ingassing is a valuable addition. The code (provided on Github) will be a wonderful resource for the community. I fully support publication of the revised manuscript.

Reviewer #1 (Remarks on code availability):

The code is accessible and clearly documented. The tutorial notebooks provide an excellent introduction to the model.

Reviewer #2 (Remarks to the Author):

Reviewer #2 (Remarks on code availability):

Code was available and able to be run.

Reviewer #3 (Remarks to the Author):

The authors were very thorough in their response to my comments and the comments of the other two reviewers. I feel like a lot of my previous comments were related to confusion in the way the text read and the way the figures were referenced. The authors have done a great job of fixing this in the revised manuscript. The revised manuscript reads very well and addresses all my previous comments related to the models and calculation parameters. The authors did a good job clarifying the calculation parameters

and I feel more confident about the use of the dimensionless parameter L in the calculations and other model assumptions. In conclusion, I believe the authors have adequately addressed all my concerns with the manuscript and the manuscript is publishable in its current state.

Minor Comments:

Figure 5: Fix the spacing between the lower two histograms so the y-axis numbers aren't cut off.

Thanks for catching that. This has now been fixed for Fig. 5.

Review of manuscript NCOMMS-25-71651: “Prolonged cooling and degassing of Apollo 17 volcanic glasses on the lunar surface”

Summary:

This manuscript constrains a three-stage decompression and cooling history for Apollo 17 volcanic glasses: in the first stage of the model, degassing occurs in response to decompression in the volcanic conduit; in the second stage, the glass beads and embayments undergo diffusive and evaporative loss of volatiles during free flight; and in the third stage of the model, the glasses undergo a final stage of volatile loss during slow cooling on the lunar surface. Volatile concentration gradients in glass beads and melt embayments are best fit by models in which the glasses undergo slow cooling and prolonged degassing on the lunar surface for timescales on the order of years. **This is a novel and interesting finding that leverages previously collected volatiles data with a sophisticated new model.**

The model is carefully constructed. The one-dimensional model for the embayment is checked against a two-dimensional model and is found to agree within 15%. The model uses appropriate parameters and constraints from the literature. In addition to the thorough description of the model provided in the manuscript, the authors state that the code will be provided on Github, which will provide the detail needed for the work to be reproduced. The model results support the conclusions of the manuscript.

We support the publication of this manuscript in Nature Communications. We provide below some suggestions for additional topics of discussion and minor corrections that could strengthen the manuscript.

Major comments:

1. A linear relationship between H₂O and pressure is assumed, but experimental evidence has shown that H₂O follows a square root relationship with pH₂O (e.g., Newcombe et al. 2017). How might the assumption of a square root relationship affect the results?
2. Su et al. (2023) suggest that S, Cl and F (which are all enriched on the surfaces of 74220 orange glass beads) are expected to ‘ingass’ (i.e., diffuse into the beads) as they travel through cool regions of the volcanic plume. The model presented here does not consider ingassing. If ingassing has occurred, the initial stages of ingassing would “flatten” the concentration gradients of H₂O, Cl, S and F, which might cause a bias in the modeled results towards models in which evaporation dominates over diffusion. Notably, this process (referred to as low-temperature rehydration) has also been observed in water concentration gradients across terrestrial melt embayments; e.g. Befus et al. (2024). A discussion of the potential effects of ingassing on the H₂O, Cl, S and F gradients would strengthen the manuscript.

Befus, Kenneth S., et al. "Rehydrated glass embayments record the cooling of a Yellowstone ignimbrite." *Geology* 52.7 (2024): 507-511.

3. If it is possible to use the embayment data to report constraints on the decompression rate of the lunar magma (P_0/t_{end}), that would be of great interest.
4. At what stage of the decompression and cooling history is post-entrapment crystallization occurring? Is it possible that vapor expansion in the conduit could have led to some adiabatic cooling prior to eruption? E.g., La Spina et al. (2015), Newcombe et al. (2020):

La Spina, G., M. Burton, and M. de'Michieli Vitturi. "Temperature evolution during magma ascent in basaltic effusive eruptions: A numerical application to Stromboli volcano." *Earth and Planetary Science Letters* 426 (2015): 89-100.

Newcombe, Megan E., et al. "Magma pressure-temperature-time paths during mafic explosive eruptions." *Frontiers in Earth Science* 8 (2020): 531911.

Minor comments:

1. After figure 1, the figure numbers within the text appear to be inaccurate.
2. Figure 3: The panels and text are small. Could this be a 2x2 tiling with the legend placed in the 4th panel?
3. Line 7: "thermal evolution of the mantle". Please clarify that you are referring to the *lunar* mantle.
4. Line 10: Please clarify "volatile-rich". Do you mean more volatile-rich than the rest of the lunar mantle? Note that scientists from different fields have different views of what constitutes "volatile-rich", so it would be helpful to provide a comparison.
5. Line 34: Please clarify what is meant by "uniform Ni concentrations that correlate with Mg". Do you mean uniform Ni within each pyroclastic deposit?
6. Line 36: Please emphasize the link between Ti content and glass colors earlier in the text. Glass colors and glass Ti-contents are used interchangeably (e.g. line 35 and 45) and this connection might not be obvious to a reader.
7. Line 39: "evidences" should be "evidence".
8. Line 56: Perhaps the significance of these glass beads should be emphasized. What is the interest in these samples compared to other lunar lithologies for "understanding lunar mantle chemistry and Moon formation"? This is briefly touched on in the abstract, but may warrant a few more sentences in the introduction.

“Pyroclastic volcanism, although volumetrically minor compared to mare volcanism, is thought to originate from deeper, more primitive sources, and volatile-rich in composition”

9. Line 73: “contains” is missing an s
10. Line 100: “volatiles content” shouldn’t be plural
11. On line 256, what is the difference between t_{inf} and t_{end} ? Please clarify.
12. Line 289: I am unable to find a cooling timescale of 500 s in Saal et al. 2008. I see modeled times of 120, 300, and 600 s.
13. Line 341: Cite the preliminary science report?
14. Line 410: “a high percentages” should be “a high percentage”
15. Line 441: “interpretating” should be “interpreting”
16. Line 519: Missing “of” between amount and time.